# Synaptic mechanisms for associative learning in the cerebellar nuclei

Robin Broersen [1,2,6], Catarina Albergaria [3,5,6], Daniela Carulli [4,6], Megan R. Carey [3,7] ✉, Cathrin B. Canto [1,2,7] ✉ & Chris I. De Zeeuw [1,2,7] ✉

Associative learning during delay eyeblink conditioning (EBC) depends on an intact cerebellum. However, the relative contribution of changes in the cerebellar nuclei to learning remains a subject of ongoing debate. In particular, little is known about the changes in synaptic inputs to cerebellar nuclei neurons that take place during EBC and how they shape the membrane potential of these neurons. Here, we probed the ability of these inputs to support associative learning in mice, and investigated structural and cell-physiological changes within the cerebellar nuclei during learning. We find that optogenetic stimulation of mossy fiber afferents to the anterior interposed nucleus (AIP) can substitute for a conditioned stimulus and is sufficient to elicit conditioned responses (CRs) that are adaptively well-timed. Further, EBC induces structural changes in mossy fiber and inhibitory inputs, but not in climbing fiber inputs, and it leads to changes in subthreshold processing of AIP neurons that correlate with conditioned eyelid movements. The changes in synaptic and spiking activity that precede the CRs allow for a decoder to distinguish trials with a CR. Our data reveal how structural and physiological modifications of synaptic inputs to cerebellar nuclei neurons can facilitate learning.

The cerebellum has a remarkable ability to learn sensorimotor associations. One well-known example of such associative learning is delay eyeblink conditioning (EBC), a Pavlovian task where well-timed conditioned eyelid closures (conditioned response; CR) arise after pairing a neutral conditioned stimulus (CS) with an aversive unconditioned stimulus (US) at a fixed interval, with both stimuli co-terminating in time[1,2]. The main inputs of the cerebellar system that convey the CS and US information have been characterized. The current working hypothesis is that the CS is at least partially conveyed to granule cells in the cerebellar cortex by mossy fiber (MF) axons originating in the pontine nuclei (PN)[3,4]. Granule cells give rise to parallel fibers and innervate Purkinje cells (PCs). Climbing fibers, derived from the inferior olive and powerfully innervating PCs, respond strongly to the US[5]. As incoming MFs and climbing fibers innervate both the cerebellar cortex and the cerebellar nuclei (CN)[6–10], both sites are strong candidates that may contribute to the plasticity mechanisms that underlie associative learning.

In the cerebellar cortex, synaptic weight changes at parallel fiber synapses onto PCs[11–13] and molecular layer interneurons[14,15], potentially in combination with intrinsic mechanisms[16] and structural changes of axonal fibers[17,18], lead predominantly to reductions in PC simple spike activity during the CS-US interval[14,19,20]. This in turn leads to disinhibition of CN neurons[21,22] and ultimately increased spiking activity of premotor and motor neurons downstream in the red nucleus and facial nucleus, respectively[23,24], thereby generating a conditioned eyelid movement.

[1]Department of Cerebellar Coordination and Cognition, Netherlands Institute for Neuroscience, Royal Netherlands Academy of Arts and Sciences, Amsterdam, The Netherlands. [2]Department of Neuroscience, Erasmus MC, Rotterdam, The Netherlands. [3]Neuroscience Program, Champalimaud Center for the Unknown, Lisbon, Portugal. [4]Laboratory for Neuroregeneration, Netherlands Institute for Neuroscience, Royal Netherlands Academy of Arts and Sciences, Amsterdam, The Netherlands. [5]Present address: University College London, Sainsbury Wellcome Centre, London, UK. [6]These authors contributed equally: Robin Broersen, Catarina Albergaria, Daniela Carulli. [7]These authors jointly supervised this work: Megan R. Carey, Cathrin B. Canto, Chris I. De Zeeuw. ✉e-mail: megan.carey@neuro.fchampalimaud.org; c.canto@nin.knaw.nl; c.dezeeuw@erasmusmc.nl

Within the complex of the CN, the anterior interposed nucleus (AIP) contains most EBC encoding neurons[21,22,25]. Manipulating their activity disrupts both the acquisition and expression of CRs[26–28]. Furthermore, CRs remain partly intact after removal of the cerebellar cortex in fully conditioned mice[29], suggesting that part of the memory is stored at the level of the nuclei[30]. However, in contrast to the synaptic plasticity mechanisms in the cerebellar cortex that contribute to eyeblink conditioning, those in the CN remain largely obscure.

Here, we investigated the role of MF synaptic inputs to CN neurons in delay EBC. We show that using optogenetic stimulation of MF terminals in the AIP as a conditioned stimulus results in well-timed eyelid movements over the course of conditioning. Moreover, during EBC, the density of inputs from excitatory MFs, as well as inhibitory inputs, is increased. Finally, in vivo whole-cell recordings of AIP neurons in awake head-fixed mice during EBC reveal the dynamics of their membrane potential ($V_m$) in relation to associative learning. Presentation of the CS induces predominantly $V_m$ depolarizations in AIP neurons, but both the timing and the directionality of the resulting eyelid movements reverse during conditioning. $V_m$ depolarizations precede the CR in conditioned mice, but follow eyelid openings in naive mice. Using linear discriminant analysis, we show that synaptic as well as spike activity can successfully inform a decoder to distinguish trials with and without a CR. Together, our findings raise the intriguing possibility that there is the capacity for sensorimotor learning and temporal coding within the CN themselves, carried by a synergy of structural and physiological changes at the level of individual nuclei neurons.

## Results

### Optogenetic stimulation of mossy fibers in the cerebellar nuclei is sufficient to induce well-timed EBC

Previous work has shown that optogenetic stimulation of MF terminals within the cerebellar cortex can act as a CS for delay EBC[31]. We asked whether optogenetic stimulation of MF terminals in the AIP could also act as a CS for EBC (Fig. 1a), and, if so, whether the learned responses could be distinguished from those driven by a cortical MF CS (from here on termed MF-CTX CS). To test this, we placed optic fiber implants just over the AIP of Thy1-ChR2/EYFP transgenic mice[32]

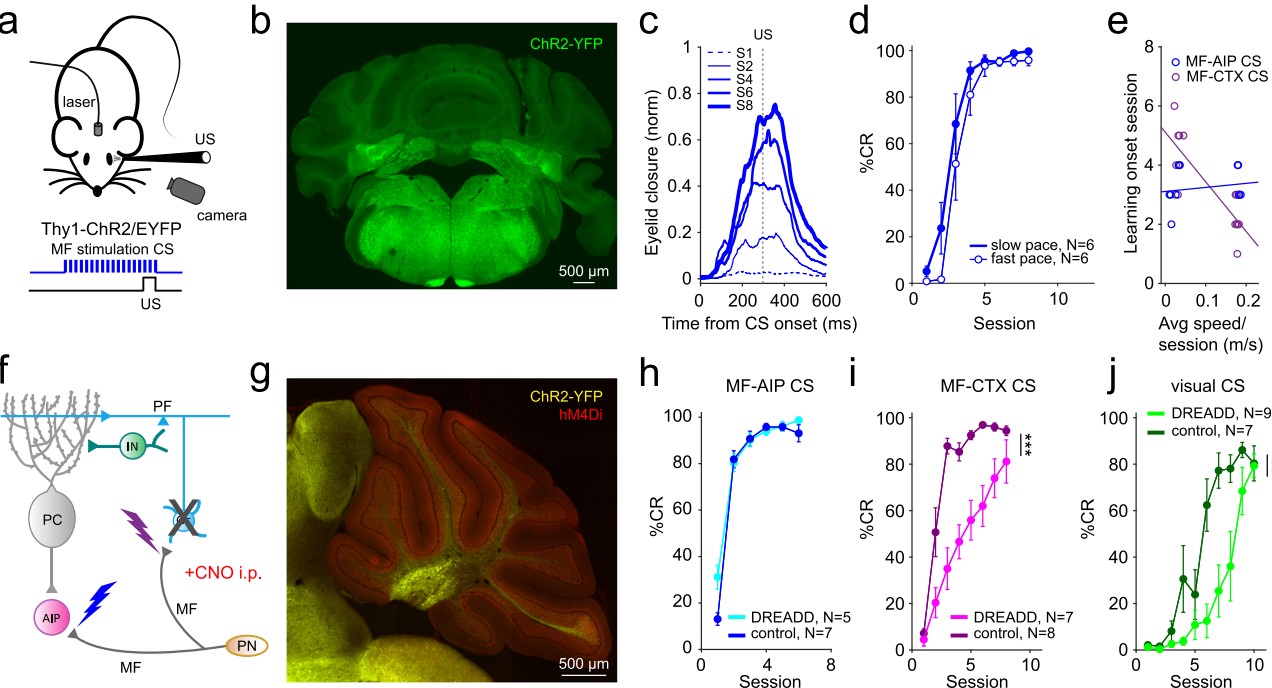

**Fig. 1 | Mice can learn to optogenetic stimulation of mossy fibers in the cerebellar nuclei as a conditioned stimulus. a** Schematic of a head-fixed mouse on a treadmill with an optic fiber implanted just above the AIP. Optogenetic stimulation of mossy fiber (MF) terminals in the AIP as a conditioned stimulus (CS) was paired with an air-puff to the ipsilateral eye (unconditioned stimulus, US). The two stimuli co-terminated for all experiments. **b** Representative histological sample indicating the placement of an optical fiber over the AIP of a Thy1-ChR2-YFP mouse expressing ChR2-YFP in MFs. ChR2-YFP expression and fiber placement was verified for all the mice. **c** Average eyelid traces of sessions from a representative animal trained to optogenetic stimulation of MF terminals in the AIP (MF-AIP CS). Each trace represents the average of CS-only trials from sessions 1, 2, 4, 6 and 8. **d** Average %CR learning curves to a MF-AIP CS for animals walking on a self-paced treadmill (slow) and those running at a fixed speed of 0.18 m/s on a motorized treadmill (fast) (main-effect of locomotion: $F_{(1,10)} = 2.688$, $P = 0.132$, main-effect of session: $F_{(2.168, 21.68)} = 61.51$, $P < 0.001$, interaction-effect locomotion x session: $F_{(7, 70)} = 0.619$, $P = 0.738$, 2-way repeated measures ANOVA). **e** Onset session of learning (average CR amplitude of session exceeds 10% of UR) plotted against the average walking speed on the treadmill of animals trained to a MF-AIP CS (average onset session 3.3 vs. 3.1, $t_{(10)} = -0.447$, $P = 0.66$, unpaired $t$ test, two-tailed; blue) and to a MF-CTX CS (average onset session 2.1 vs. 4.6, $t_{(12)} = 5.376$, $P < 0.001$, unpaired $t$ test, two-tailed; purple). Lines represent linear fits (MF-CTX: $N = 14$, slope = −16.8, $P < 0.001$; MF-AIP:

$N = 12$, slope = 1.4, $P = 0.564$). **f** Cerebellar circuit diagram with lightning bolts representing the sites of optogenetic stimulation: mossy fiber terminals in the lobule simplex (purple) and MF terminals in the AIP (blue). MF, mossy fiber; Gc, granule cell; PC, Purkinje cell; IN, interneuron; PF, parallel fiber; AIP, anterior interpositus nuclei. **g** Fluorescence image of a parasagittal cerebellar section of a MF-ChR2-gc-DREADD mouse, indicating ChR2-YFP expression in MFs (yellow fluorescence), and hM4Di receptors (DREADDs) selectively in granule cells (red fluorescence). Similar expression was found in 11 other mice. **h–j** Average %CR learning curves of granule cell-expressing DREADD mice (lighter colors) and respective non-DREADD-expressing controls (darker colors), injected with CNO prior to each training session. **h** Animals trained to a MF-AIP CS (main-effect of DREADD: $F_{(1,10)} = 0.762$, $P = 0.403$, main-effect of session: $F_{(2.348,23.48)} = 128.4$, $P < 0.001$, interaction-effect DREADD x session: $F_{(5,50)} = 2.267$, $P = 0.062$, 2-way repeated measures ANOVA). **i**, Animals trained to a MF-CTX CS (main-effect of DREADD: $F_{(1,13)} = 9.186$, *** $P < 0.001$, main-effect of session: $F_{(2.770,32.45)} = 31.3$, $P < 0.001$, interaction-effect DREADD x session: $F_{(7,82)} = 2.665$, $P = 0.016$, linear mixed models). **j** Animals trained to a natural visual CS (main-effect of DREADD: $F_{(1,14)} = 10.81$, ** $P = 0.005$, main-effect of session: $F_{(2.394,27.13)} = 18.74$, $P < 0.001$, interaction-effect DREADD x session: $F_{(9,102)} = 3.133$, $P = 0.002$, linear mixed models). Data in (**d**, **h**, **i**, **j**) is shown as mean ± SEM. Source data are provided as a Source Data file.

expressing Channelrhodopsin-2 (ChR2) in cerebellar MFs[33] (from here on termed MF-ChR2 mice). MF afferents to AIP prominently express ChR2-YFP in these mice (Fig. 1b). We used 350 ms low-power (sub-threshold for movement) laser stimulation as a CS, paired with a 50 ms air-puff US delivered to the ipsilateral eye, so that the two stimuli co-terminated. Similar to learning to a MF-CTX CS[31], mice learned to make CRs in response to optogenetic stimulation of MF terminals in the AIP (from here on termed MF-AIP CS, Fig. 1c).

The conditioned responses to a MF-AIP CS increased gradually during learning and were well-timed in that eyelid closures peaked at the end of the CS-US inter-stimulus interval (ISI) of 300 ms (Fig. 1c, d). However, unlike learning to either a sensory CS or the learning driven by MFs in the cerebellar cortex[31], learning to a MF-AIP CS was not enhanced by locomotor activity ($P > 0.05$, Fig. 1d), and there was no difference in how quickly animals learned on a slow self-paced vs. fast motorized treadmill (AIP: average onset session 3.3 vs. 3.1, $P > 0.05$; CTX: average onset session 2.1 vs. 4.6, $P < 0.001$; Fig. 1e). The differential effects of locomotion on EBC for MF-AIP vs. MF-CTX CSs parallel the previous finding that locomotor activity preferentially enhanced eyelid closures evoked through the cerebellar cortex and not the nuclei[31] and they suggest that different mechanisms may govern learning through the two pathways.

We further examined the specificity of optogenetic stimulation of MF terminals in the AIP vs. CTX by combining optogenetic MF CS stimulation in AIP with the use of inhibitory Designer Receptors Exclusively Activated by Designer Drugs (DREADDs)[34,35] expressed in cerebellar granule cells, the targets of MFs in the cerebellar cortex (Fig. 1f), under the control of the Gabra6 promoter[36]. These mice expressed ChR2 in MFs as well as the inhibitory DREADD hM4Di in granule cells throughout the cerebellar cortex, as confirmed by immunohistochemistry (Fig. 1g; mice from here on termed MF-ChR2-gc-DREADD). Administration of the DREADD agonist clozapine-N-oxide (CNO) intraperitoneally to MF-ChR2-gc-DREADD mice prior to each training session had no effect on learning to an optogenetic MF-AIP CS, compared to CNO administration in non-DREADD expressing MF-ChR2 littermate controls ($P > 0.05$; Fig. 1h). In contrast, CNO impaired learning to either a MF-CTX CS ($P < 0.001$; Fig. 1i) or a visual CS ($P < 0.01$; Fig. 1j) in MF-ChR2-gc-DREADD mice. Together, the differential effects of locomotor activity and granule cell inhibition on learning to a MF-AIP vs. MF-CTX CS confirm the selective stimulation of the MF-CN pathway.

In all of the experiments presented so far, the CRs generated by a MF-AIP CS appeared to be properly timed, with the peak amplitude appearing just before the expected air-puff, similar to a cortical MF CS[31] or a visual CS (Supplementary Fig. 1a-d). However, these experiments do not speak to the capacity to learn to a longer ISI, nor to the ability to adjust the timing of CRs adaptively. To test this, we designed an experiment that allowed us to look at the ability of individual mice to adapt their CRs to changing ISIs (Supplementary Fig. 1e). Mice subjected to MF-AIP CS (Supplementary Fig. 1f), as well as those subjected to MF-CTX CS (Supplementary Fig. 1g), could learn to a long (500 ms) ISI, and adaptively time their CRs to different ISIs (Supplementary Fig. 1h). Moreover, prolonging the ISI, when training to a MF-AIP CS, resulted in smaller average amplitudes of the CR ($Z = 10$, $P = 0.034$, $N = 4$ mice, Wilcoxon, one-tailed; Supplementary Fig. 1f), similar to previous studies on the impact of longer ISI protocols in both wild types and mutants with cerebellar cortical defects[37,38].

## Conditioning induces structural changes in pontine mossy fiber input to the AIP

Our finding that optogenetic activation of MF inputs in the AIP can serve as a reliable CS suggests that plasticity mechanisms in the AIP can directly contribute to learning and storage of the associative memory trace. We therefore asked what structural changes in inputs to the AIP occur during normal associative learning to a sensory CS. Mice were

subjected to 10 days of conditioning ('cond') or pseudo-conditioning ('pseudo', i.e., CS and US at random rather than fixed ISIs, preventing learning; Supplementary Fig. 2a), and we subsequently evaluated the number of excitatory and inhibitory synapses (Fig. 2). All quantifications to investigate structural changes during learning were done by an investigator blind to the experimental group. The vast majority of excitatory synapses in the CN belong to terminals of MFs and climbing fibers, whereas inhibitory synapses belong mainly to PCs. Presynaptic excitatory terminals can be detected by using antibodies against the vesicular glutamate transporter 1 (VGLUT1) or 2 (VGLUT2). Neurons in the pontine nuclei (PN) sending MFs to the cerebellum predominantly express VGLUT1 mRNA, whereas neurons in the inferior olive, from which the climbing fibers originate, express VGLUT2 mRNA[39,40].

We found a significantly increased density of VGLUT1+ puncta in the AIP of conditioned mice when compared to pseudo-conditioned mice ($P < 0.05$; Fig. 2a–c). No significant difference was detected between the ipsilateral and contralateral side in conditioned mice (ipsi vs. contra: $t_{(10)} = 1.876$, $P = 0.090$, unpaired $t$ test, two-tailed, $N = 11$). We found a positive correlation between VGLUT1 density in the ipsilateral AIP and the mean CR amplitude of the last conditioning session ($r^2 = 0.35$, $P = 0.056$, Pearson's correlation, $N = 11$), suggesting that the number of VGLUT1+ terminals in the ipsilateral AIP coincides with a larger conditioned eyelid response (Fig. 2e). No correlation was found for the contralateral AIP or for both sides pooled (contra AIP: $r^2 = 0.074$, $P = 0.42$; bilateral AIP: $r^2 = 0.20$, $P = 0.17$, Pearson's correlation; Fig. 2f) or in pseudo-conditioned mice (contra AIP: $r^2 = 0.026$, $P = 0.62$; ipsi AIP: $r^2 = 0.087$, $P = 0.35$; bilateral AIP: $r^2 = 0.032$, $P = 0.58$, Pearson's correlation). As for VGLUT2+ terminals, no significant change was found in the AIP after conditioning ($P > 0.05$; Fig. 2d) and no significant correlation was observed between VGLUT2 density and CR amplitude (contra AIP: $r^2 = 0.025$, $P = 0.84$; ipsi AIP: $r^2 = 0.0090$, $P = 0.90$; bilateral AIP: $r^2 = 0.0012$, $P = 0.97$, Pearson's correlation, $N = 4$).

To confirm that the increases in VGLUT1+ puncta represent changes in pontocerebellar MFs, we overexpressed yellow fluorescent protein (YFP) in MFs by injecting mice with AAV9-hSyn-hChR2(H134R)-EYFP bilaterally in the PN[41]. Mice were subsequently conditioned or pseudo-conditioned with a unilateral US on the left side (Supplementary Fig. 2a) and we evaluated the density of YFP+ axonal varicosities in the AIP (number/MF length; Fig. 2g–i). The vast majority of YFP+ varicosities were also VGLUT1+ (>95%), indicating that we quantified largely VGLUT1 + MF axons. We found a significant increase in the amount of varicosities in the AIP in conditioned mice compared to pseudo-conditioned mice ($P < 0.01$; Fig. 2g–i). Again, no significant difference was detected between the ipsilateral and contralateral side in conditioned mice ($t_{(3)} = 0.9401$, $P = 0.416$, unpaired $t$ test, two-tailed, $N = 3$).

We also investigated whether the increase in number of MF synapses in the AIP of conditioned mice is accompanied by changes in MF terminals in the eyeblink area of the cerebellar cortex. MF terminals (also known as rosettes) consist of multiple glutamatergic synapses on granule cell dendrites. Rosettes were visualized thanks to YFP-reactivity in AAV-injected mice. The vast majority (>95%) of YFP+ rosettes contained VGLUT1 (Fig. 2j). When we quantified the size of YFP + rosettes in the ipsilateral and contralateral side of conditioned mice, we again did not detect any difference (D = 0.08, $P = 0.30$, Kolmogorov-Smirnov test, two-tailed, $N = 5$). Moreover, no difference was present between conditioned and pseudo-conditioned mice (D = 0.0016, $P > 0.99$, Kolmogorov-Smirnov test, cond: $n = 613$ YFP+ rosettes, $N = 5$ mice, pseudo: $n = 761$, $N = 4$; Fig. 2k).

To investigate whether the inhibitory input to the AIP also changes at the structural level during conditioning, we quantified the number of gephyrin+ clusters in putative excitatory AIP neurons (i.e., neurons with a soma size >250 μm$^2$)[42,43]. Gephyrin is the core scaffolding protein in the inhibitory post-synaptic density, which anchors glycine receptors and GABAA receptors at postsynaptic sites[44,45]. The

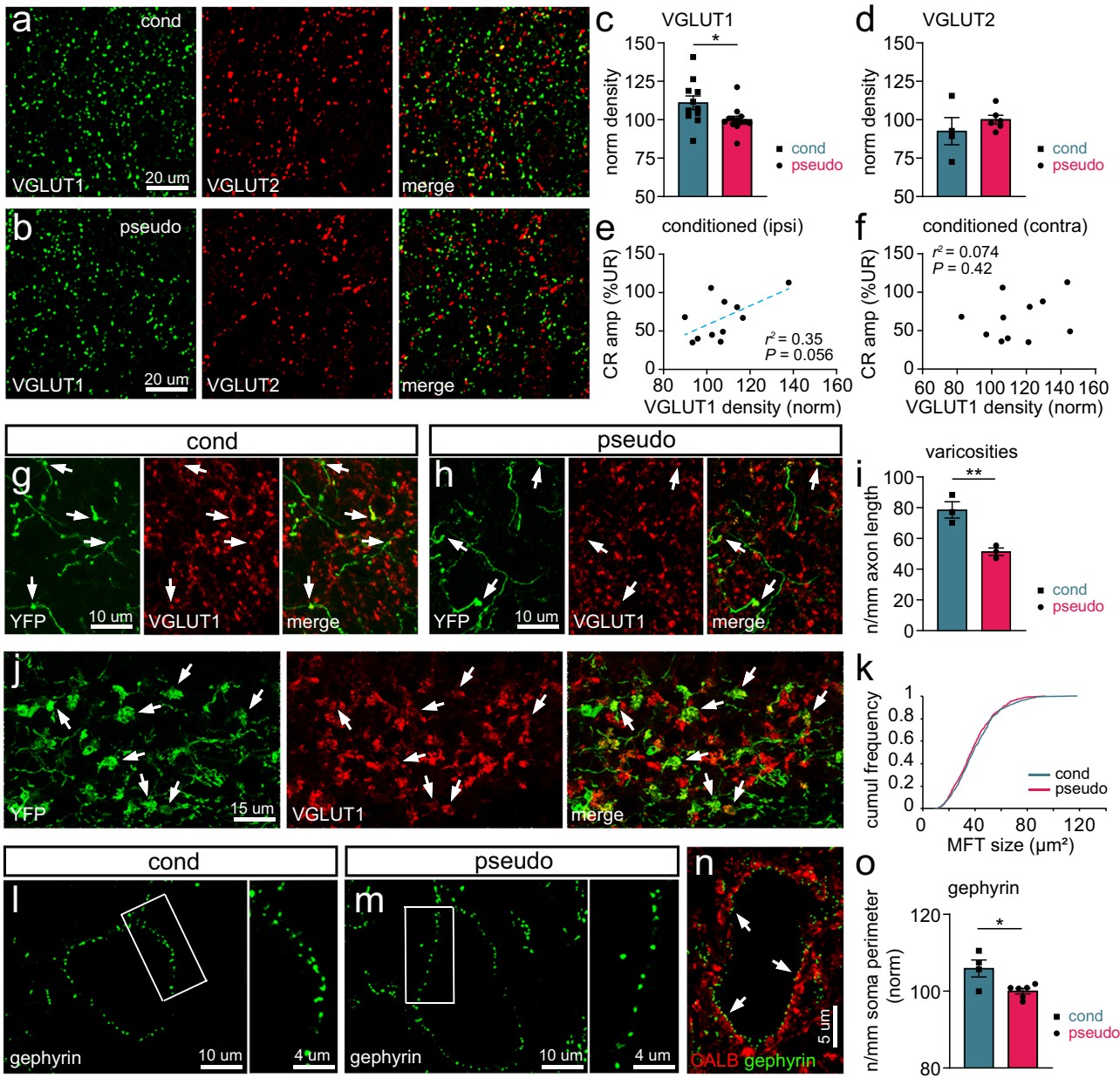

**Fig. 2 | Learning-induced changes in VGLUT1+ mossy fiber terminals and gephyrin+ inhibitory terminals.** Expression of VGLUT1 and VGLUT2 in the AIP of a representative conditioned (cond; **a**) and pseudo-conditioned (pseudo; **b**) mouse. **c** The density of VGLUT1+ puncta in the AIP is higher in conditioned mice compared to pseudo mice ($t_{(22)} = 2.352$, * $P = 0.028$, unpaired $t$ test, two-tailed, cond: $N = 11$, pseudo: $N = 13$ mice). **d** The density of VGLUT2+ puncta in the AIP is unchanged after conditioning ($t_{(8)} = 0.9532$, $P = 0.37$, cond: $N = 4$, pseudo: $N = 6$ mice). Density of VGLUT1+ puncta in the ipsilateral (**e**) or contralateral (**f**) AIP plotted against the CR amplitude (%UR) at the last training session for conditioned animals. VGLUT1 density correlates positively with the CR amplitude in the ipsilateral (**e**) but not in the contralateral (**f**) side. Representative pictures of YFP+ mossy fibers and varicosities, which mostly contain VGLUT1 (arrows), in conditioned (**g**) and pseudo (**h**) mice. **i** The number of varicosities/mm axon length is increased in conditioned mice

when compared to pseudo mice (bilateral AIP in pseudo vs. conditioned mice: $t_{(4)} = 4.689$, ** $P = 0.0094$, unpaired $t$ test, two-tailed, cond: $N = 3$, pseudo: $N = 3$). **j** Representative pictures of mossy fiber terminals (MFT; or rosettes), labeled by YFP and for the vast majority containing VGLUT1 (arrows), in the EBC area of the cerebellar cortex. **k** The size of YFP + /VGLUT1+ MFTs in conditioned mice ($N = 5$, $n = 613$) is not different from that of pseudo mice ($N = 4$, $n = 761$). Expression of gephyrin in the AIP of conditioned (**l**) and pseudo (**m**) mice. **n** The vast majority of gephyrin+ clusters in a putative glutamatergic AIP neuron are apposed to CALB+ terminals (arrows). **o** Conditioned mice show a significantly higher density of gephyrin+ clusters when compared to pseudo mice ($t_{(8)} = 3.042$, * $P = 0.016$, unpaired $t$ test, two-tailed, cond: $N = 4$, pseudo: $N = 6$). Data is shown as mean ± SEM. Source data are provided as a Source Data file.

density of gephyrin puncta in the AIP was significantly increased in conditioned mice ($P < 0.05$; Fig. 2l–o) as compared to pseudo-conditioned mice (data from both sides were pooled as gephyrin puncta densities were comparable, $t_{(2)} = 0.241$, $P = 0.832$, paired $t$ test, two-tailed, $N = 4$). To determine whether the increase in gephyrin+ puncta can represent an increased input from PC terminals, we quantified the percentage of gephyrin+ puncta that were apposed to

calbindin-positive boutons (Fig. 2n). Given that calbindin is expressed by PCs as well as by climbing fibers[46,47], but only PC terminals are inhibitory, gephyrin+ puncta that are apposed to calbindin-positive boutons represent inhibitory connections from PCs. We found that in both conditioned and pseudo-conditioned mice, around 98% of gephyrin+ puncta were apposed to PC terminals (conditioned mice: 97.4 ± 0.54%; pseudo-conditioned mice: 98.15 ± 0.35%; $U = 480.5$,

$P = 0.551$, Mann–Whitney $U$ test, two-tailed, cond: $N = 3$, pseudo: $N = 3$), indicating that the increase in gephyrin puncta following conditioning probably largely reflects a change in PC transmission.

Together, these data indicate that terminals of pontocerebellar MFs to the AIP as well as inhibitory PC axons increase bilaterally over the course of associative learning, whereas MF inputs to the main eyeblink region in the cerebellar cortex remain unchanged.

## Optogenetic stimulation of the pontine mossy fibers after EBC results predominantly in spike facilitation in AIP neurons

Following conditioning, the predominant response of AIP neurons is spike facilitation induced by the CS, with the spike rate increasing during learning[22]. We next set out to investigate whether activity in MFs from the PN can lead to activity changes in the AIP after conditioning. We made juxtasomal loose-patch recordings from AIP neurons of awake mice that had been injected with AAVs expressing the excitatory opsin Chronos in the PN (Supplementary Fig. 3a, b). Our recordings at the end of conditioning day 10 showed that 75% of the neurons showed changes in spike activity during optogenetic stimulation of the PN with 465 nm (blue) light (Supplementary Fig. 3c). Blue light stimulation resulted in spike facilitation with a latency shorter than 10 ms in 62% of the neurons, suggestive of direct excitatory MF activation.

To confirm that this spike facilitation was due to a direct MF connection and not due to disinhibition in the MF−granule cell−PC pathway, we subsequently made whole-cell recordings of CN neurons in anesthetized Gabra6::ChR2EYFP mice[48] that selectively express ChR2 in granule cells (Supplementary Fig. 4a, b). Most (i.e., 13 out of 15) of the recorded AIP neurons showed mean decreases rather than mean increases in spiking in response to granule cell-ChR2 stimulation (Supplementary Fig. 4c, d), indicating that activation of the MF−granule cell−PC pathway is unlikely to result in spike facilitation in CN neurons and that the facilitation outcomes highlighted above could indeed result from direct MF to CN neuron stimulation.

## Optogenetic stimulation of the pontine nuclei leads to larger eyelid movements after conditioning

We next tested whether eyelid movements induced by optogenetic stimulation of PN neurons change over conditioning. Mice were injected with AAVs expressing Chronos and/or eArch3.0 in the PN. Co-injecting the two opsins allowed us to both activate and inhibit PN neurons using light stimulation of different wavelengths. Next, we conditioned mice to a visual CS for 10 days (Supplementary Fig. 2b). We recorded eyelid movements on day 0 (D0), day 5 (D5) and day 10 (D10) that were generated in response to optogenetic stimulation at a consistent light intensity across days (Supplementary Fig. 3d). In response to blue light stimulation, we found that conditioned mice made increased short-latency eyelid closures on D10 compared to D0 (area under the curve (AUC) 0–150 ms: $P < 0.05$; Supplementary Fig. 3e), while the peak-time of eyelid closures remained unchanged ($86.8 \pm 4.52$ ms after light onset, $F_{(1.254, 3.761)} = 2.308$, $P = 0.214$, one-way repeated-measures ANOVA). Eyelid closures were not observed during amber light stimulation inhibiting PN neurons (AUC 0–150 ms: $P > 0.05$; Supplementary Fig. 3f), ruling out that this was a non-specific reflexive eyelid closure to onset of the light. Together, these results suggest that increased VGLUT1+ MFs to the AIP in conditioned mice may support larger eyelid movements after conditioning.

## Membrane potential dynamics during CS-evoked eyelid movements

The data presented above raise the question as to how the changes in excitatory MF inputs and inhibitory inputs shape the membrane potential ($V_m$) of AIP neurons over the course of EBC. We therefore recorded from AIP neurons in whole-cell configuration during EBC in awake head-fixed mice on a treadmill (Fig. 3a). A green LED flash (CS)

signaled the arrival of a corneal air-puff (US) into the left eye with an ISI of 250 ms. We subjected naive and fully conditioned mice, the latter of which were conditioned for at least 10 days (Supplementary Fig. 2c), to a single session of in vivo whole-cell recordings in current-clamp mode targeting the AIP (Fig. 3a). Correct targeting was validated by inspection of a fluorescent tracer (Fig. 3b), electrical microstimulation and CS-evoked spike response patterns[22,25]. All mice were presented with alternating CS-US paired and CS or US unpaired trials during recordings. As expected, conditioned mice showed significantly more CRs than naive mice and their CRs showed higher amplitudes ($P < 0.001$; Fig. 3c). Interestingly, in 66.7% of the recordings, naive mice showed eyelid openings (EO) in response to the CS, movements directly opposite to classical CRs ($P < 0.05$). Thus, naive and conditioned mice show directionally opposite eyelid responses to the CS, providing an opportunity to study the membrane potential of AIP neurons during stimulus-evoked eyelid movements towards different directions.

To investigate the physiological mechanisms underlying these opposite movements, simultaneous recordings of the $V_m$ in current-clamp mode and eyelid movements were acquired ($n = 15$ neurons per group; Fig. 3a, Supplementary Table 1). In vitro recorded AIP neurons can be classified into multiple cell-types[43,49], but classification based on in vivo electrophysiological properties alone appeared problematic[50,51]. AIP neurons showed a diversity of $V_m$ responses to paired and unpaired CS and US stimuli, as well as responses during spontaneous voluntary eyelid movements (Fig. 3d, e, arrowheads). The population response including all neurons showed a transient $V_m$ depolarization in naive mice and a more sustained depolarization in conditioned mice, coinciding with directionally opposite eyelid movements (Fig. 3f). Receiver Operating Characteristic (ROC) analysis revealed that the percentage of CS-responding neurons was 73.3% in naive mice ($n = 11$) and 80% in conditioned mice ($n = 12$; Fig. 3g). Most CS-responders showed $V_m$ depolarizations after CS onset (naive, 53.3% of all neurons; conditioned, 66.7%; Fig. 3d, e). Since these depolarizations could be the result of excitatory input or reductions in tonic inhibitory input from PCs, we classified these neurons as 'CS-activated' (Supplementary Movie 1). A subset of neurons showed $V_m$ hyperpolarizations and were classified as 'CS-suppressed' (naive, 20%; conditioned, 13.3%; Supplementary Fig. 5a–c; Supplementary Movie 2). The remaining neurons did not show any $V_m$ responses evoked by the CS ('CS-no response', naive, 26.7%; conditioned, 20%; Supplementary Fig. 5d).

Overall, the distribution of CS responses was comparable between groups ($\chi^2_{(2)} = 1.083$, $P = 0.582$, chi-square test, $n = 15$ per group) (Fig. 3g). Since CS-activated AIP neurons were most prevalent, and eyelid movements are thought to be driven by increased spike activity in excitatory projection neurons to downstream facial and oculomotor nuclei[23], we focused on CS-activated AIP neurons in our subsequent analysis. $V_m$ depolarizations in CS-activated neurons in naive and conditioned neurons were similar in amplitude ($P > 0.05$), peak time ($P > 0.05$) and area during the last 200 ms before the US ($P > 0.05$; Fig. 3h, i). However, the area during the last 40 ms before the US was significantly lower in naive neurons compared to conditioned neurons ($P < 0.05$; Fig. 3i, gray area). This significant difference remained when excluding the most positive and negative value in the naive group ($U = 10$, $P = 0.031$, Mann–Whitney $U$ test, two-tailed). This shows that CS-evoked depolarizations underlying directionally opposite eyelid movements are transient in CS-activated naive neurons, but sustained in conditioned neurons.

## $V_m$ depolarizations reflect CRs in a subset of conditioned AIP neurons

To further investigate how conditioned eyelid movements are encoded, we separated $V_m$ responses in trials with a CR (CR trials) and trials with no-eyelid response (NR trials; Fig. 4a). The initial rising phase of depolarizations appeared to be comparable between CR and NR trials,

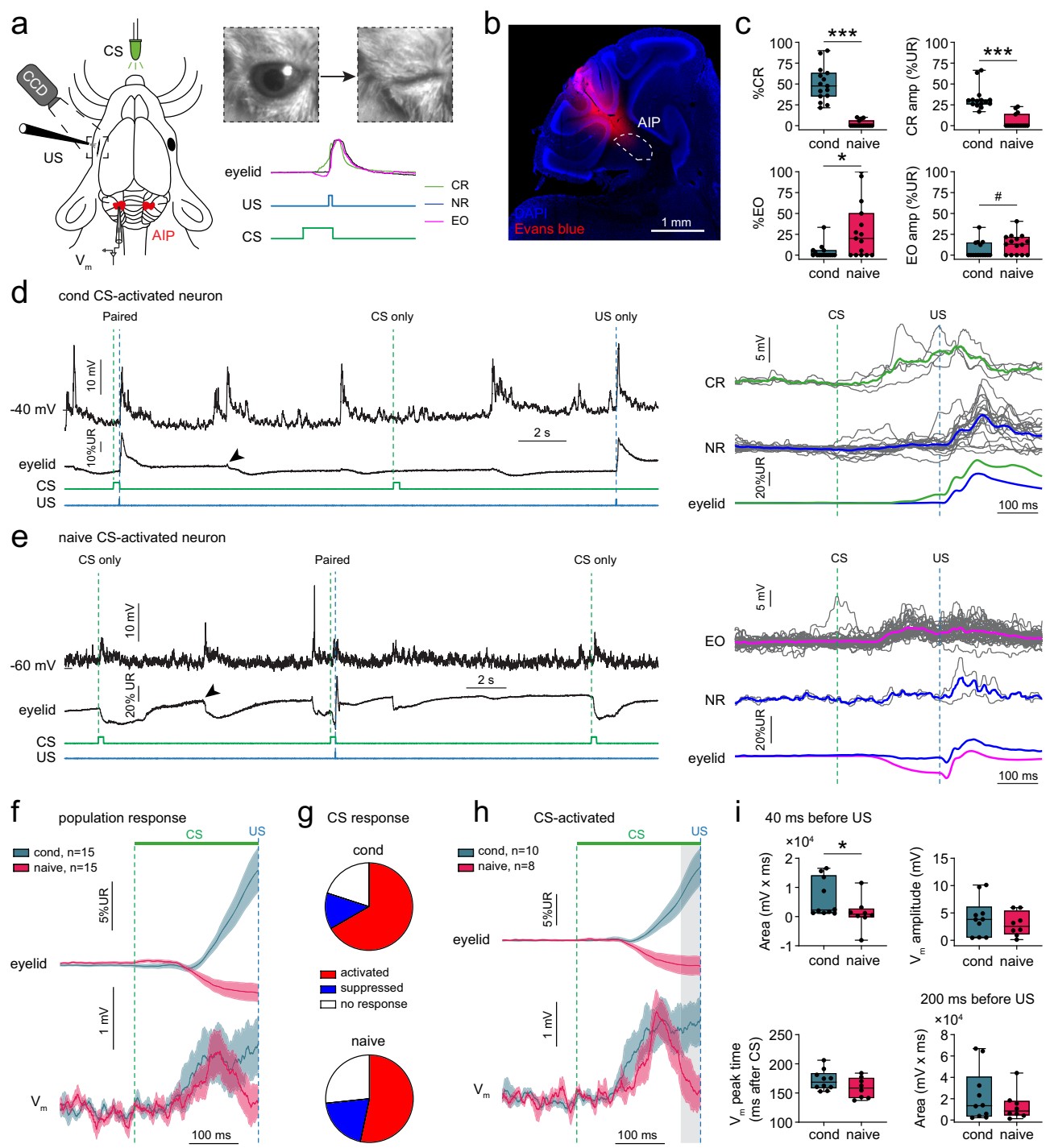

but we hypothesized that in some neurons higher sustained depolarizations on CR trials occurred during the last 50 ms before the US (Fig. 4a). We performed ROC and correlation analysis and found that only 33.3% of conditioned CS-activated responders showed CR dependency (Fig. 4b), whereas 66.7% of neurons responded similarly on CR and NR trials (Fig. 4c, Supplementary Fig. 6a). CR-dependent neurons showed larger $V_m$ amplitudes ($P < 0.05$) and areas during the last 50 ms before the US on CR versus NR trials ($P < 0.05$; Fig. 4b, gray area, 4d). Furthermore, on a group level, both $V_m$ amplitude and area correlated with eyelid movements ($V_m$ amplitude: $r = 0.435$, $P < 0.05$; area: $r = 0.545$, $P < 0.01$; Supplementary Fig. 6b). As expected, for CR-independent neurons, $V_m$ amplitude and area did not correlate with eyelid movements ($V_m$ amplitude: $r = 0.14$, $P > 0.05$; area: $r = 0.035$, $P > 0.05$; Supplementary Fig. 6c). Among CS-suppressed neurons we

did not find a clear relationship between $V_m$ hyperpolarizing responses and eyelid movements ($V_m$ amplitude: $r = 0.157$, $P > 0.751$; area: $r = 0.172$, $P = 0.6$; Supplementary Fig. 5e, f). In summary, CS processing can be widely observed in the membrane potential of many AIP neurons and part of them also show depolarizations in a CR-dependent fashion.

**$V_m$ depolarizations reflect eyelid openings in naive AIP neurons**
All naive CS-activated neurons for which we recorded both EO and NR trials showed EO dependency (Fig. 4e, f). $V_m$ amplitudes, but not areas during the last 50 ms before the US were larger on EO trials than on NR trials ($V_m$ amplitude: $P < 0.01$; area: $P > 0.05$; Fig. 4g). On a group level both the $V_m$ amplitude and area correlated significantly with eyelid amplitude ($V_m$ amplitude: $r = 0.462$, $P < 0.001$; area: $r = 0.413$,

**Fig. 3 | Membrane potential responses during the conditioned stimulus in cerebellar nuclei neurons. a** Schematic of in vivo whole-cell recordings of head-fixed mice during EBC. Eyelid responses to CS and US stimuli are recorded using a high-speed camera. Upward deflections in the eyelid trace indicate eyelid closures, downward deflections indicate eyelid openings. CR: conditioned response, EO: eyelid opening, NR: no response, CS: conditioned stimulus, US: unconditioned stimulus, AIP: anterior interpositus nuclei, CCD: charge-coupled device camera. **b** Recordings were targeted at the AIP. Electrode track is visualized by coating the electrode with Evans blue dye. **c** Performance of conditioned and naive mice during the recordings (cond: $n = 15$ recordings from 12 mice; naive: $n = 15$ recordings from 11 mice), showing higher %CR and CR amplitudes during recordings from conditioned mice (%CR: $U = 0$, *** $P < 0.001$, Mann-Whitney $U$ test, two-tailed; CR amplitude: $U = 3$, *** $P < 0.001$, Mann–Whitney $U$ test, two-tailed), but higher %EO and EO amplitudes during recordings from naive mice ($U = 58$, * $P = 0.015$, Mann–Whitney $U$ test, two-tailed; EO amplitude: $U = 69$, # $P = 0.056$, Mann–Whitney $U$ test, two-tailed). **d-e** Example traces of a conditioned and a naive CS-activated neuron. **d** Conditioned AIP neuron showing $V_m$ depolarizations during paired, CS-only and US-only trials, but also during spontaneous eyelid movements (arrowhead). Larger depolarizations are seen on CR trials compared to NR trials. **e** Naive AIP neuron showing $V_m$ depolarizations during paired and CS-only trials, and during spontaneous eyelid movements (arrowhead). Larger depolarizations are seen on EO trials compared to NR trials. **f** Population eyelid and $V_m$ response averages across all neurons show an eyelid closure and sustained $V_m$ depolarization for conditioned mice and an eyelid opening coinciding with a transient depolarization for naive mice. **g** Distribution of CS-evoked $V_m$ responses for both groups, showing in both cases that the predominant response classification is CS-activation. **h** Eyelid and $V_m$ response averages for CS-activated AIP neurons. Gray area indicates the last 40 ms before the US used calculation of area in (i). **i**, Graphs showing quantifications of $V_m$ responses shown in (**h**). No differences in amplitude ($t_{(16)} = 0.857$, $P = 0.404$, unpaired $t$ test, two-tailed, $n = 10$ conditioned/8 naive), peak time ($t_{(16)} = 1.567$, $P = 0.137$, unpaired $t$ test, two-tailed) and area during the last 200 ms before the US ($U = 35$, $P = 0.697$, Mann–Whitney $U$ test, two-tailed). The area during the last 40 ms before the US is significantly higher in conditioned than in naive neurons ($U = 17$, * $P = 0.043$, Mann–Whitney $U$ test, two-tailed). Boxplots show median and 25th–75th percentiles, whiskers show minimum and maximum values. All other graphs and traces show mean ± SEM. Source data are provided as a Source Data file.

$P < 0.001$; Supplementary Fig. 6d). However, CS-suppressed naive neurons did not show larger $V_m$ hyperpolarizations on EO trials compared to NR trials ($V_m$ amplitude: $t_{(2)} = 2.934$, $P = 0.099$; area: $t_{(2)} = 1.429$, $P = 0.289$, paired $t$ test, two-tailed, $n = 3$ neurons). Neither did we find a correlation between $V_m$ hyperpolarizations and eyelid movements ($V_m$ amplitude: $r = 0.084$, $P = 0.927$; area: $r = 0.104$, $P = 0.726$; Supplementary Fig. 5g, h). Together, our data suggest that AIP neurons widely show CS-evoked $V_m$ responses reflecting conditioned and unconditioned CS-eyelid movements, but with a varying degree of correlation with the movement.

## Conditioning, but not pseudo-conditioning, leads to CS-evoked whole-body movements

One possibility for the varying degree of correlation with eyelid movements is that a subset of our recorded AIP neurons processes body movements other than the eyelid. Indeed, cerebellar rostral AIP neurons have previously been shown to drive coordinated whole-body movements[52]. To investigate whether conditioning leads to CS-evoked body movements, we acquired video recordings of mice subjected to conditioning or pseudo-conditioning and calculated the sum of frame-by-frame pixel changes as a measure of macroscopic body movements (movement index; Supplementary Fig. 7a, Supplementary Movie 3). During the first training session (D1), before the emergence of CRs, mice showed a short-latency response to CS onset (Supplementary Fig. 7b, black arrowhead) followed by a moderate increase in movement. After 10 days of training (D10), conditioned mice developed increased CS-evoked movements (Supplementary Fig. 7b, red arrowhead), whereas pseudo mice did not (AUC CS period: $P < 0.01$; Supplementary Fig. 7d). The short-latency CS-responses did not change during training and remained comparable between groups (main-effect of training day: $F_{(1,13)} = 0.161$, $P = 0.694$, main-effect of group: $F_{(1,13)} = 0.876$, $P = 0.366$, interaction-effect training day vs. group: $F_{(1,13)} = 0.951$, $P = 0.347$, 2-way repeated measures ANOVA). Both conditioned and pseudo mice showed a higher variability in baseline movements on D10 versus D1, but this effect was comparable between groups ($P < 0.01$; Supplementary Fig. 7e). The magnitude of CS-evoked movements correlated significantly with CR amplitude on a trial-by-trial basis for 55.6% (5 out of 9) of conditioned mice.

The US alone also evoked movements, but neither conditioning nor pseudo-conditioning changed the magnitude of these movements (AUC US period: $P > 0.05$; Supplementary Fig. 7c, f). These results indicate that eyeblink conditioning concurrently leads to generalized body movements, overall correlated with eyelid movements. Therefore, we cannot discount the possibility that at least part of the $V_m$ responses in AIP neurons represent movements of other body parts.

## $V_m$ responses precede conditioned eyelid movements, but follow CS-evoked eyelid openings

We reasoned that if AIP neurons play a causative role in generating CRs, $V_m$ responses should precede the onset of CRs. In contrast, unconditioned EOs in naive mice may well be the result of cerebral cortical activity[53,54] and their onset would not necessarily precede the onset of EOs. To test this, we quantified the onset of $V_m$ and eyelid responses blindly on a trial-by-trial basis (Fig. 4h). We found that CRs started $147.3 \pm 7.6$ ms after CS onset and that EOs started $99.4 \pm 5.2$ ms after CS onset ($n = 11$ conditioned/8 naive neurons; Fig. 4j). Conditioned and naive $V_m$ responses started $126.9 \pm 6.3$ ms and $123.7 \pm 4.7$ ms after CS onset, respectively. On a trial-by-trial basis, $V_m$ responses started $22.88 \pm 13.72$ ms earlier than CRs in conditioned neurons, whereas $V_m$ responses started $25.23 \pm 6.92$ ms later than EOs in naive neurons ($P < 0.05$; Fig. 4i, j). These data indicate that the timing underlying eyelid movements becomes temporally precise to precede the conditioned eyelid movement. The response in naive AIP neurons could represent an efference copy of the command or feedback signal related to the EO, but likely does not represent an early signal that generates the EO itself.

## $V_m$ responses follow spontaneous eyelid movements

Since we found that depolarizing $V_m$ responses may play a causative role in CR generation, we asked how this compares to spontaneous eyelid openings or closures occurring in between stimuli. Spontaneous eyelid movements likely do not coincide with movements in other parts of the body, thereby providing us with a measure of the level of eyelid specificity of $V_m$ responses. ROC analysis revealed that $V_m$ responses during spontaneous eyelid movements occurred in almost all conditioned (91.67%) and more than half of naive (58.33%) neurons, thereby confirming that we were largely targeting eyelid-specific AIP neurons (Supplementary Fig. 8a). 75% of all neurons showed responses only during openings or closures but not during both. Furthermore, openings (52.38%) and closures (47.62%) were comparably represented amongst the responders. There was no difference in $V_m$ responses between conditioned and naive mice ($V_m$ amplitude: $F_{(3,11)} = 0.308$, $P = 0.819$, one-way ANOVA, two-tailed; eyelid amplitude $F_{(3,3.564)} = 1.098$, $P = 0.456$, Brown-Forsythe ANOVA, two-tailed, $n = 6$ conditioned/7 naive neurons), so we pooled the data from both groups of mice. $V_m$ depolarizing responses to spontaneous eyelid openings and closures, which were comparable in terms of eyelid amplitude (opening $15.55 \pm 1.28$ %UR, closure: $19.7 \pm 5.95$ %UR, $P > 0.05$; Supplementary Fig. 8b), also had comparable $V_m$ amplitudes (opening: $7.12 \pm 1.83$ mV, closure: $5.6 \pm 1.18$ mV, $P > 0.05$; Supplementary Fig. 8c). Across all pooled occurrences, the amplitudes of $V_m$ depolarizations and eyelid movements did not

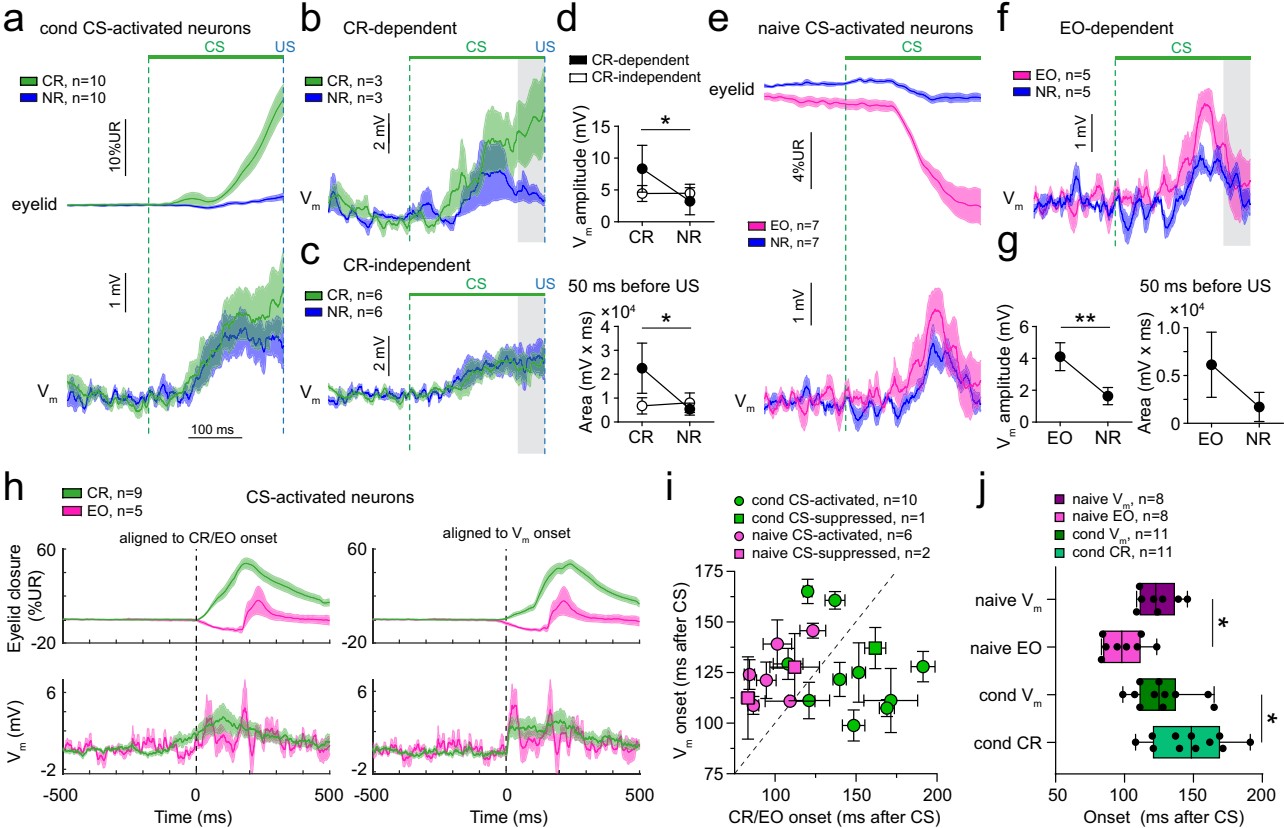

**Fig. 4 | Relationship between membrane potential responses in CS-activated neurons and eyelid movements. a** Average eyelid and $V_m$ traces on conditioned response (CR) and no response (NR) trials of all CS-activated neurons in conditioned mice. **b** Average $V_m$ traces on CR and NR trials for neurons that show different responses on both types of trials (CR-dependent). Most prominent differences occur during the last 50 ms before the US (gray area). **c** Same as (**b**), but for neurons that show similar responses on CR trials compared to NR trials (CR-independent). **d** Graphs comparing the $V_m$ amplitude and the area within the gray area in (**b**) and (**c**). $V_m$ amplitude and area during the last 50 ms before the US are increased on CR trials for CR-dependent, but not for CR-independent neurons (amplitude: main-effect of eyelid response: $F_{(1,7)} = 6.734$, $P = 0.036$, interaction-effect eyelid response vs. CR dependency: $F_{(1,7)} = 6.972$, $P = 0.033$, CR vs. NR for CR-dependent group: * $P = 0.03$, 2-way repeated measures ANOVA with Šídák's multiple comparisons test; area: main-effect of eyelid response: $F_{(1,7)} = 7.064$, $P = 0.033$, interaction-effect eyelid response vs. CR dependency: $F_{(1,7)} = 9.567$, $P = 0.018$, CR vs. NR for CR-dependent group: * $P = 0.019$, 2-way repeated measures ANOVA with Šídák's multiple comparisons test, $n = 3$ CR-dep/6 CR-indep). **e** Average eyelid and $V_m$ traces on eyelid opening (EO) and no response (NR) trials of all CS-activated neurons in naive mice. **f** Average $V_m$ traces on EO and NR trials for neurons that

show different responses on both types of trials (EO-dependent). Gray area indicates 50 ms before start of the US. **g** Graphs comparing the $V_m$ amplitude and area during the 50 ms before the US, showing that the amplitude, but not the area, is significantly increased on EO trials compared to NR trials (amplitude: $t_{(3)} = 6.258$, ** $P = 0.008$; area: $t_{(3)} = 1.679$, $P = 0.192$, paired $t$ test, two-tailed, $n = 4$ neurons). **h** Eyelid and $V_m$ traces on CR or EO trials for CS-activated neurons recorded in conditioned or naive mice, respectively, aligned to the onset of the CR/EO (left panels) or the onset of the $V_m$ response (right panels). Traces include a combination of CS and US responses since the CS-US delay is 250 ms and CS-only and paired trials are pooled. **i** Onset of $V_m$ response versus onset of CR or EO for all conditioned and naive neurons, respectively. On average, $V_m$ responses start earlier than CRs, but later than EOs. **j** Onsets of $V_m$ responses and EO or CR responses relative to the onset of the CS. As shown in (**i**), $V_m$ responses in naive mice start after EOs, but $V_m$ responses start before CRs in conditioned mice ($F_{(3,34)} = 8.964$, $P < 0.001$, $V_m$ vs. CR: * $P = 0.044$, $V_m$ vs. EO: * $P = 0.039$, one-way ANOVA with Šídák's multiple comparisons test). Boxplots show median and 25th–75th percentiles, whiskers show minimum and maximum values. Data in all other graphs is shown as mean ± SEM. Source data are provided as a Source Data file.

significantly correlate, neither for openings ($r = 0.066$, $P = 0.418$) nor closures ($r = 0.028$, $P = 0.791$; Supplementary Fig. 8e).

Overall, spontaneous eyelid movements started $26.14 \pm 8.83$ ms earlier than $V_m$ depolarizations ($t_{(9)} = 2.995$, $P = 0.008$, one-sample $t$ test, one-tailed, $n = 10$ neurons) and this was consistent for both openings and closures ($P > 0.05$; Supplementary Fig. 8d). For neurons that showed $V_m$ hyperpolarizations, we observed that eyelid movements started $34.91 \pm 11.14$ ms earlier than $V_m$ hyperpolarizations ($t_{(3)} = 3.135$, $P = 0.026$, one-sample $t$ test, one-tailed), a lag similar to $V_m$ depolarizations ($P > 0.05$; Supplementary Fig. 8f). These lags were almost identical to timings observed during EOs, where EOs started $25.23 \pm 6.92$ ms earlier than $V_m$ responses (spontaneous vs. EO lags: $P > 0.05$). In contrast, as discussed earlier, CRs started $22.88 \pm 13.72$ ms after $V_m$ responses (Supplementary Fig. 8f).

## Spike activity during conditioned response trials ramps up during ongoing CRs

To investigate how $V_m$ responses are translated into spike activity during EBC we performed juxtasomal loose-patch recordings in conditioned mice, while recording CR and NR trials. Out of 112 recorded neurons (average spike rate $72.97 \pm 3.08$ Hz, range 3.6–165.3 Hz), 50% of the neurons responded with significant changes in spike rate to the CS. Of these responders 76.8% showed spike facilitation and 23.2% showed spike suppression (Supplementary Fig. 9a). These percentages are in good agreement with the distribution of CS-evoked $V_m$ responses (Fig. 3g) as well as previously published data[22]. CS-responding neurons were located in both the ipsilateral (52.7% of $n = 55$ neurons) and contralateral (33.3% of $n = 33$ neurons) hemisphere relative to the US. Neurons that responded with spike facilitation to the CS showed on average significantly increased spike rates during CR

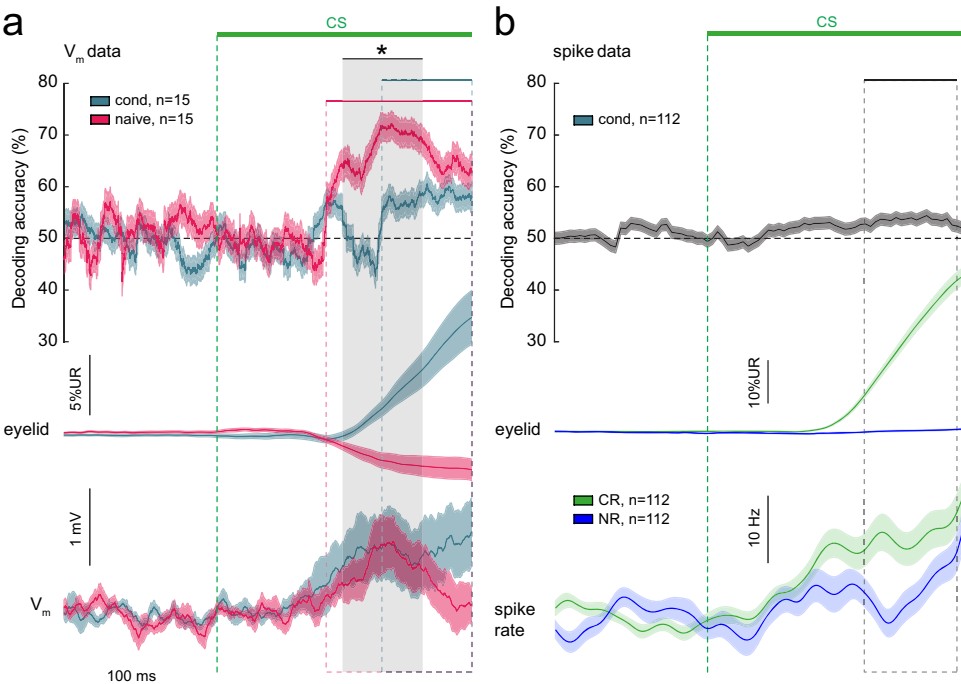

**Fig. 5 | Decoding of eyelid movements based on conditioned stimulus-evoked neuronal activity. a** Decoding of CR versus NR trials based on $V_m$ data from all conditioned neurons and decoding of EO versus NR trials based on $V_m$ data from all naive neurons. Decoding was significantly above chance for both groups, but during different time periods relative to the CS-US interval. Time periods of significant decoding are indicated with the colored horizontal bars and dotted lines for each group. Gray area indicates the period during which decoding based on naive neurons performed significantly better than decoding based on conditioned neurons. Eyelid and $V_m$ traces are visualized as reference. **b** Decoding of CR versus NR trials based on juxtasomal spike data from all conditioned neurons. Decoding was significantly above chance within the period indicated with the horizontal bar and dotted line, coinciding with the upward sloping CR and largest differences in the average spike rates (lower panels). Data is shown as mean ± SEM. *$P < 0.05$.

trials (CR: 48.22 ± 5.78 Hz, NR: 31.12 ± 4.95 Hz, $P < 0.001$), while the timing of peak spike rate was comparable among trials (CR: 202.6 ± 9.98 ms, NR: 176.2 ± 11.43 ms, $P > 0.05$; Supplementary Fig. 9b-d). Similar to $V_m$ recordings, average spike rate traces revealed that not only on CR, but also on NR trials, transient increases in spike rates occur. Yet again, only during the late phase of the CS-US interval these differences become prominent, temporally coinciding with the sustained phase of $V_m$ depolarizations in CS-activated neurons. We correlated the number of spikes during the last 125 ms of the CS-US period on CR trials with eyelid amplitudes across spike facilitation neurons and found a significant correlation between CR amplitude and spike count ($r = 0.218$, $P < 0.001$; Supplementary Fig. 9e). We also tested whether spike count during the first 125 ms correlates with CR amplitude, which it did ($r = 0.139$, $P < 0.001$, Spearman, two-tailed, $n = 648$ trials in $N = 30$ neurons), although this correlation was weaker than that for spikes during the second half of the CS-US interval ($Z = -2.299$, $P = 0.022$, Fisher's Z-transformation). Together, these data argue for a close translation from subthreshold to spike activity, where spikes during the ongoing CR, coinciding with the sustained phase of $V_m$ depolarizations, are best correlated with the size of the conditioned eyelid response.

**Decoding of eyelid behavior from $V_m$ responses and spike data**
To determine whether CS-evoked eyelid responses can be decoded from trial-by-trial $V_m$ population data, we performed linear discriminant analysis (LDA) on spike-removed $V_m$ traces, using a tenfold cross-validation procedure to train and test a classifier. Conditioned ($n = 199$ CR trials and $n = 265$ NR trials in 15 neurons) and naive traces ($n = 113$ EO trials and $n = 214$ NR trials in 15 neurons) were used to decode CR vs. NR trials, and EO vs. NR trials, respectively. This analysis retains the temporal dynamics of the decoding accuracy across different phases of CS-US trials. We found that the decoder

performed significantly above chance level for both naive and conditioned traces during the CS-US interval (Fig. 5a). Significant above-chance decoding was observed starting 107.1 and 161.7 ms after the onset of the CS and lasted until US onset, for naive and conditioned traces, respectively. Furthermore, decoding accuracy from naive traces was significantly higher than that from conditioned traces between 123.1 and 201.7 ms after the CS (Fig. 5a, gray area). We also performed LDA on pooled spiking data from the juxtasomally recorded CN neurons in conditioned mice for comparison ($n = 1444$ CR trials and $n = 414$ NR trials in 112 neurons). The decoder performed significantly above chance between 155 and 245 ms after the CS (Fig. 5b). Peak decoding accuracies were 72.2% for naive $V_m$ traces, 60.6% for conditioned $V_m$ traces and 54.4% for conditioned spiking data. These peak decoding accuracies occurred at 170.6 ms for naive $V_m$, 205.9 ms for conditioned $V_m$ and 220 ms after CS onset for conditioned spike data.

We also asked how decoding accuracy changes as a function of group size. Decoding was performed on pooled trial data from groups of 15 to 3 neurons, where for each group we randomly selected a unique subset of neurons for 29 iterations. As expected, for both groups the decoder performed best with the maximal group size and accuracy declined with smaller group sizes (Supplementary Fig. 10a, b). The time the decoder performed significantly above chance declined significantly as a function of group size for both groups ($P < 0.05$; Supplementary Fig. 10c). The reverse trend was observed for variation in decoding accuracy, which increased with decreasing group sizes ($P < 0.001$), but not for mean decoding accuracy during the period of above-chance significance ($P > 0.05$; Supplementary Fig. 10d, e). In summary, while $V_m$ and spike data from both groups can sufficiently inform a decoder to distinguish trial types, decoding based on naive traces leads to the highest decoding accuracies covering the longest time span.

### Encoding of the unconditioned stimulus in AIP neurons

We next asked what the $V_m$ dynamics are underlying the US, a 10 ms air-puff directed at the left eye. We found that all recorded conditioned and naive CN neurons showed significant $V_m$ responses to the US, with $V_m$ depolarizations as the most common response (cond: 80% US-activated, 13.3% US-suppressed, 6.7% US activated/suppressed; naive: 86.7% US-activated, 13.3% US-suppressed; Supplementary Fig. 11a, b). Typically, if neurons showed $V_m$ depolarizations after the CS, they also did after the US, although some neurons without CS-response did show US responses. Since there was significant variation in US-evoked $V_m$ depolarizations as well as UR amplitudes between groups, our dataset was insufficient to detect a difference in $V_m$ area (effect size d = 0.325, Df = 19, power = 0.175; Supplementary Fig. 11c, d). However, comparing CR vs. NR trials on paired trials within the conditioned group revealed that preceding CRs led to reduced US-evoked $V_m$ depolarizations ($P < 0.05$; Supplementary Fig. 11e, f). A similar effect of a preceding EO on US-evoked $V_m$ depolarizations was not observed for naive neurons ($P > 0.05$; Supplementary Fig. 11g, h). Thus, a preceding CR influences $V_m$ encoding of the US, consistent with the protective function of the conditioned eyelid closure.

## Discussion

In this study we shed light on the synaptic and neuronal coding mechanisms during cerebellar learning in the AIP, the part of the cerebellar nuclei that is crucial for both the acquisition and expression of delay EBC behavior[26–28]. We show that stimulating MF afferents to AIP as a CS can effectively drive learning, with conditioned mice exhibiting well-timed CRs that are adaptive to changing CS-US delays. Chemogenetic inhibition of cerebellar granule cells selectively impairs learning to a cortical optogenetic CS, but leaves learning to a MF-AIP CS intact. Moreover, learning with MF-AIP stimulation as a CS is not modulated by locomotor activity, in contrast to learning with cortical MF stimulation[31]. Taken together, these results strongly suggest that there is a capacity for well-timed learning within the AIP, without the need for MFs to convey CS information to the cerebellar cortex[55–57].

We further show that learning leads to structural changes in both excitatory MF and inhibitory inputs to the AIP, while MF input to the granule cell layer remains unchanged. At the same time we find that optogenetic stimulation of the PN neurons, the main source of MF input to the cerebellum, elicits larger eyelid movements after conditioning. Last but not least, by performing in vivo whole-cell recordings in awake behaving mice during EBC, we show how these learning-induced changes shape the $V_m$ dynamics of AIP neurons. We find that conditioned neurons sustain a depolarized $V_m$ state until US onset, while naive mice show transient $V_m$ depolarizations. After conditioning the $V_m$ response becomes temporally precise in that it starts earlier than eyelid closures, away from the initial delay between $V_m$ response and eyelid openings in naive mice. This indicates that conditioned AIP neurons could play a role in initiating the CR, whereas naive neurons likely process an efference copy of the eyelid opening with movements initiated by other brain areas[53,54]. This does not rule out the possibility that AIP activity modulates the ongoing CR or EO, which is likely since eyelid movements and $V_m$ responses are well-correlated in a substantial subset of AIP neurons.

Juxtasomal spike recordings predominantly show increased spike responses during the second half of the CS-US interval, which temporally coincides with the sustained $V_m$ depolarization as well as the upward ramping phase of CRs. Both $V_m$ and spike data are sufficient to inform a decoder to separate movement from no-movement trials during similar periods of the CS-US interval, despite the fact that a considerable proportion of AIP neurons shows CR-independent $V_m$ responses and some show no detectable CS-responses. This response heterogeneity may reflect a combination of the cellular diversity of the CN[42,43,49–51] and differences in input patterns and activity. AIP neurons may also differentially encode aspects of the conditioned movement,

or even movements of multiple body parts during defensive behavior[52]. Nevertheless, an open question remains to what extent CS responses correlate with CN cell types and what are the precise contributions of inhibitory and excitatory inputs to the $V_m$ and spike response. Here we chose to prioritize current-clamp recordings to measure how AIP neurons behave during EBC without experimenter interference. This approach did not allow us to distinguish inhibitory versus excitatory input, as possible during voltage-clamp recordings.

There is broad consensus that CRs emerge as the result of well-timed pauses in PC activity in response to the CS[14,19], where plasticity in the AIP would be under the control of PC inputs[58]. However, our results indicate that at least part of the memory trace may be generated and stored directly at the level of the nuclei[30], at least in part through structural modifications of VGLUT1 + MF and inhibitory inputs. This would be in agreement with the observation that many AIP neurons gradually increase their spike rates over the course of learning, coinciding with larger conditioned eyelid movements[22]. Besides changes in the number of inputs, other forms of plasticity may take place in parallel. For instance, changes in synaptic ultrastructure[59] and transmission, intrinsic excitability[60], postsynaptic re-localization of receptors or changes in activity from inputs[14], may also take place[61]. This could occur under the control of perineuronal nets harboring chemo-attractive or repulsive molecules[62]. Our recordings show the sum total of these changes as they occur during learning, thereby providing a unique insight into how the output of the cerebellum is conveyed to downstream motor nuclei before and after learning.

One has to be aware of several potential caveats in our study, e.g., CNO is known to have potential off-target effects. Although we used a concentration that is well within the range typically used for hM4Di receptors[35] and we implemented proper controls in the form of non-DREADD-expressing animals injected with CNO, the potential effects of repeated injections of CNO have not been well documented. Furthermore, the precise location of our whole-cell recordings could not be determined post-hoc, as the relatively short duration of recordings (149 s median) did not allow for reliable neurobiotin filling. In response to this limitation we performed surface-to-depth measurements, microstimulation, fluorescent tracer injections as well as validation of spike activity to ensure that we targeted the AIP, although this does not ascertain that all recorded neurons were from AIP. We further observed that mice develop whole-body movements, which are correlated with the eyelid movement after conditioning, but not pseudo-conditioning. Activity of AIP neurons has been shown to correlate with control of multiple body parts[52] and thus our estimation of eyelid-related activity in the AIP may be overestimated. However, it should be noted that similar $V_m$ activity was also present during spontaneous eyelid movements, which occur isolated from other body movements, arguing in favor of eyelid-specificity. In the same line of reasoning, one could hypothesize that optogenetic stimulation of PN neurons may lead to increased whole-body movements besides the eyelid. To determine whether this is the case and to determine to what extent eyelid-encoding AIP neurons are also involved in other functions requires a follow-up study.

Nevertheless, our findings shed light on the longstanding debate regarding the relative contributions of the cerebellar cortex and nuclei for cerebellum-dependent motor learning. Lesion and reversible inactivation studies have suggested that these two areas may encode different features of learning, with plasticity at MF-AIP synapses mediating the expression of motor responses, while cortical cells regulate their gain and timing[21,28,63–72]. One of the most influential models in the field posits that temporally specific plasticity in the cerebellar cortex leads, over learning, to well-timed pausing of PCs, and, consequently, a downstream disinhibition of AIP neurons[63]. Indeed, AIP neurons, which are like PCs highly intrinsically active[22,50], are sensitive to the differential impact of synchronized activity of PCs during behavior[73]. Our findings demonstrate how a combination of

structural and physiological changes at the level of MF inputs to the CN may work together with cerebellar cortical plasticity mechanisms to facilitate efficient associative learning.

## Methods

### Animals and ethics statement

All animal experimental procedures were conducted in accordance with the institutional animal welfare committees of the Erasmus Medical Center, the Royal Dutch Academy of Arts and Sciences, the Champalimaud Center for the Unknown or the Direcçãoo Geral de Veterinária (Ref. No. 0421/000/000/2015). All experiments adhered to the European guidelines for the care and use of laboratory animals (Council Directive 86/6009/EEC). Male and female C57Bl/6 J mice (4–14 weeks of age, from institutional colony and from Janvier Laboratories) were used for all experiments, except for in vivo whole-cell recordings, where we used only male mice. All mice were socially housed with *ad libitum* access to food and water. Mice for optogenetic stimulation of the mossy fibers (Fig. 1, Supplementary Fig. 1) were kept on a reverse light cycle (12:12 h light/dark) so that all experiments were performed during the dark period while mice were more active. Mice used for the remaining experiments were kept on a normal light cycle (12:12 h light/dark) and experiments were performed at day time.

### Transgenic mouse lines

The Thy1-ChR2-YFP mouse line (B6.Cg-Tg(Thy1-COP4/EYFP)18Gfng/J)[32] was obtained from The Jackson Laboratory (stock number: 007612). The MF-ChR2- gc-DREADD mouse line was obtained in two steps. First, Gabra6-Cre mice (B6.D2-Tg(Gabra6-cre)B1Lfr/Mmucd)[36] obtained from MMRRC (stock number: 015966-UCD) were crossed with R26-LSL-Gi-DREADD mice (B6.129-$Gt(ROSA)26Sor^{tm1(CAG-CHRM4*,-mCitrine)Ute}$/J)[74] obtained from The Jackson Laboratory (stock number: 026219). Second, Gabra6-Cre::DREADD animals were crossed with Thy1-ChR2-YFP mice to obtain Gabra6-Cre::Thy1-ChR2::DREADD mice. Gabra6-Cre::ChR2(H134R)-EYFP animals were obtained by crossing Gabra6-Cre mice (B6.129P2-Gabra6$^{tm2(cre)Wwis}$/Mmucd)[48] with Ai32 mice (B6.Cg-$Gt(ROSA)26Sor^{tm32(CAG-COP4*H134R/EYFP)Hze}$/J)[75] obtained from The Jackson Laboratory (stock number: 024109).

### Pedestal/head plate surgery

Mice were anesthetized with isoflurane (4–5% induction, 0.5–2% maintenance; in 0.2 L/min $O_2$ and 0.2 L/min air) and placed in a stereotaxic frame (David Kopf Instruments) where their body temperature was kept at 37 °C using a feedback-controlled heating-pad. Eyes were covered with antimicrobial ointment (Terra-Cortril, Pfizer) to prevent drying. Skin covering the skull was shaved and a mid-sagittal incision of approximately 1 cm was made to expose the skull bone, after which local anesthesia was applied on the skin around the incision (10% Xylocaine, AstraZeneca). Primer (Optibond All-In-One, Kerr) was applied on the bone and treated with UV light. A 6 x 3 x 5.7 mm aluminum block ('pedestal'), or a head plate for MF-ChR2 experiments, was attached to a stereotaxic arm and attached to the skull bone with dental acrylic (Flowline, Heraeus Kulzer or Super Bond, C&B). Skin edges were attached to the dental acrylic using tissue glue (Histoacryl, Aesculap). Post-operative analgesia was given subcutaneously with Meloxicam (2 mg/kg Metacam, Boehringer Ingelheim Animal Health) and mice were placed under an infrared light while recovering from anesthesia. Mice recovered for at least two days following surgery before subsequent experiments.

### Viral injections

Mice were prepared for surgery as described above. A longer mid-sagittal incision was made until the edge of the occipital bone. A stereotaxic arm was used to mark the location of entry to reach the lateral pontine nuclei (PN) (AP -3.8 to -4 mm, ML 1 mm, DV 5.5 to 5.8 mm rel. to Bregma), comparable to previously used coordinates[18]. Two small craniotomies were made using a dental drill (Foredom Drill K1070-2E, Blackstone Industries) and the dura was carefully removed using a fine forceps (Dumont #5). For optogenetic stimulation of the PN, mice were injected with AAV1/Syn-Chronos-GFP (UNC Vector Core, #AV6550) or co-injected with AAV2/hsyn-eArch3.0-EYFP (UNC Vector Core, #AV5229B). For tracing of fluorescently labeled MFs in the AIP and cerebellar cortex, mice were injected with AAV9-hSyn-hChR2(H134R)-EYFP (Addgene, #26973-AAV9). The Chronos and eArch3.0 viruses were diluted 4x and a total of 1 μL virus solution was injected in each hemisphere. The ChR2-EYFP virus was not diluted and 220 nL was injected in each hemisphere. In both cases we used borosilicate glass pipettes connected to a Nanoject II or Nanoject III Programmable Nanoliter Injector (1–3 μL/min, Drummond Scientific). Injections of 69 nL were made every 1 min and after the last injection the pipette was left in place for 2–10 min to allow diffusion and was then retracted.

### Optic fiber implants

For MF-ChR2 experiments, unilateral optic fibers (Ø200 μm, 0.53 NA, Doric lenses) were lowered into the brain and positioned at the cortical eyelid region (AP -5.7 mm, ML + 1.9 mm, DV 1.5 mm rel. to Bregma) or just over the AIP (AP -6 mm, ML + 1.7 mm, DV 2.1 mm rel. to Bregma)[76]. The implants were fixed into place using dental cement (Super Bond, C&B). Mice were subsequently monitored during 2 days of recovery. For the remaining experiments, bilateral optic fibers (Ø200 μm,0.39 NA, FT200UMT, Thorlabs) connected to ceramic zirconia ferrules (MM-FER2007C-2300, Precision Fiber Products) were lowered into the brain through the same injection track until a depth of 5 mm to enable light delivery to the PN. Fibers were attached to the skull bone using primer (Optibond All-In-One, Kerr) and dental cement (Flowline, Heraeus Kulzer). Dental cement was painted using silver conductive paint (RS Components) to reduce the escape of optogenetic light. Mice were allowed to recover at least 2 weeks before subsequent experiments.

### Craniotomy surgery for in vivo electrophysiology

Mice for awake whole-cell recordings were prepared for surgery as described above. Eyes were protected and kept moist using eye drops (Duodrops, Ceva Santé Animale). Hairs in the neck were removed and an incision was made in the skin covering the neck muscles, after which local anesthesia was applied (10% Xylocaine, AstraZeneca). Neck muscles were removed to expose the occipital bone and two square-shaped bilateral craniotomies were made using a dental drill. Position of craniotomy was above Crus II to access the CN in a 43° angle. A thin layer of UV-curing primer (Optibond All-In-One, Kerr) was applied around the craniotomies and a recording bath was made using dental acrylic (Flowline and Charisma, Heraeus Kulzer). The recording bath was filled with 0.9% NaCl solution to keep the dura moist. Skin and muscles surrounding the bath were attached to the dental cement using tissue glue (Histoacryl, Aesculap). The dura in the craniotomy was carefully removed using a needle tip (30 G x ½ inch, Microlance, BD) and fine forceps (Dumont #7 curved). Directly after removing the dura, the recording bath was cleaned with 0.9% NaCl solution and filled with a low viscosity silicone elastomer sealant (Kwik-cast, World Precision Instruments) to protect the brain tissue from the air. Mice were given post-operative analgesia subcutaneously (Metacam 2 mg/kg) and were placed under an infrared light during recovery. Mice were closely monitored for at least two hours before starting the recording session. Gabra6-Cre::ChR2-EYFP mice for anesthetized whole-cell recordings were subjected to a similar craniotomy surgery, but received an intra-peritoneal injection of 75 mg/kg ketamine and 12 mg/kg xylazine instead of isoflurane anesthesia. The depth of anesthesia was ensured by monitoring the heartbeat, whisker movements and reflexes to a tail-pinch. Additional ketamine-xylazine mixture (10% of initial dose) was administered if

necessary. Gabra6-Cre::ChR2-EYFP were subjected to whole-cell recordings immediately following the craniotomy surgery.

## Eyeblink conditioning habituation

Mice for MF-ChR2 experiments were trained on an experimental setup previously described[31]. Head-fixed mice were habituated for >3 days to walk at the target speed on the motorized treadmill until they walked without signs of distress. Speed of the treadmill was controlled by a DC motor with an encoder (Maxon). For all other experiments mice were handled and habituated for 3–5 days with increasing durations (15–60 min). Habituations were done in sound- and light-isolating boxes (Neurasmus) equipped with a cylindrical, non-motorized treadmill and a horizontal bar for head fixation[71]. Eyelid movements were measured using magnet distance measuring technique (MDMT)[77] or using a high-speed camera[22]. In order to place a magnet under the eye for MDMT mice were shortly anesthetized with isoflurane (5% in 0.5 L/min $O_2$ and 0.5 L/min air). Hairs under the left eyelid were removed under isoflurane anesthesia on day 2. From day 3 onward a 1.5 x 0.5 x 0.7 mm neodymium magnet (Supermagnete) was placed under the left eye using super glue (Bison International), after which they were placed on the treadmill and woke up in the dark environment of the box. Eyelid movements were recorded through a magnetic sensor that was placed above the left eye. During the habituation session, 20 CS trials and 2 US trials with an inter-trial-interval (ITI) between 10–12 ± 2 sec were presented to familiarize the mouse with the stimuli. The CS consisted of a 260 ms light flash delivered through a green/blue LED placed ~7 cm in front of the mouse. The US consisted of a 10 ms corneal air-puff (35–40 psi) delivered through a P20 pipette tip positioned ~5 mm from the left eye. The US airflow was controlled by a Milli-Pulse Pressure Injector (MPPI-3, Applied Scientific Instrumentation). CS and US stimuli were identical in camera boxes, but no magnet or magnet sensor was used. Mice also did not undergo isoflurane anesthesia in these boxes.

## Eyeblink conditioning acquisition

Mice for MF-ChR2 delay conditioning experiments were subjected to 6 to 10 days of acquisition sessions, consisting of the presentation of 90% CS-US paired trials and 10% CS-only trials. Each session consisted of 110 trials, separated by a randomized ITI of 10–15 s. In each trial, the inter-stimulus interval (ISI) between the CS and US was 300 ms, 500 ms for the timing experiments, and both stimuli always co-terminated. The US was an air-puff (30–50 psi, 50 ms) controlled by a Picospritzer (Parker) and delivered via a 27 G needle positioned 0.5 cm away from the cornea of the right eye, positioned so that it induced a full eye blink. The visual CS was a white light LED positioned 2 cm directly in front of the mouse. For all other experiments acquisition trainings lasted 5–10 days and consisted of one training session daily. A session was comprised of 200 CS-US (paired) trials, 20 CS trials and 20 US trials, presented as a 20x repeated block of 1US-10paired-1CS. Trials followed a standardized format of 0–500 ms baseline, 500–760 ms CS (paired and CS trials), 750–760 ms US (paired and US trials), 760–2000 ms post-stimuli period. MDMT data was captured at 1017.26 Hz using custom-written LabVIEW software (National Instruments)[77]. Trainings were started at least 10 min after placing mice in the box to allow full recovery of isoflurane anesthesia. Trials were initiated only if the eye was >75% opened.

## CS-US timing experiments

Mice were trained with a long ISI of 500 ms for 10 consecutive sessions, then switched to a shorter ISI of 200 ms for the duration of 3 sessions, and finally switched back to 3 sessions of 500 ms ISI. The CS and US always co-terminated. At the end of each block of sessions with a particular ISI we performed a test session where 50% of the trials were CS-only. These trials were included in our timing analysis.

## Granule cell DREADD experiments

Gabra6-Cre::Thy1-ChR2::DREADD mice received intra-peritoneal injections of CNO (5 mg/kg; Tocris Bioscience, #4936) 60–90 min before starting eyeblink experiments. Specific DREADD expression in cerebellar granule cells was confirmed histologically. Since it is difficult to ensure complete cell-type specificity in BAC transgenic Cre lines, key experiments were replicated in another granule cell-specific Cre line[48] with nonoverlapping patterns of expression outside of granule cells.

## Mice for quantification of structural changes

Mice used for the quantification of VGLUT1, VGLUT2 and Gephyrin were conditioned or pseudo-conditioned in multiple batches. Each batch consisted of age and sex-balanced conditioned and pseudo-conditioned mice. All mice were subjected to 10 days of conditioning or pseudo-conditioning, where the latter group received CS and US at random rather than fixed ISIs, which prevented learning.

## Eyeblink conditioning during electrophysiology

Mice for in vivo juxtasomal and whole-cell recordings were first trained in boxes using MDMT, after which they were transferred to an electrophysiology setup equipped with a camera. Mice were then trained for at least 3 more days on this setup before performing recording sessions, to ensure they reached maximum performance. A 250 fps CCD camera (scA640-120gc, Basler) was directed at the left eye, which was illuminated with an infrared light. Camera frames were transformed into a data vector based on pixel thresholding within a region of interest overlaying the eye and the eyelid. The resulting data was sampled at 2441 Hz. Stimuli were triggered by custom-made software utilizing TDT System 3 (Tucker Davis Technologies) and NI-PXI (National Instruments) processors. During the US the TTL onset was corrected for the delay between TTL onset and air-puff arrival at the cornea (13 ms). To avoid distortion of the camera signal caused by reflection of the infrared light from the whiskers, a thin layer of waterproof black mascara was applied on the left whiskers before mice were head-fixed. Timings and parameters of stimuli were identical to those used during the acquisition training sessions.

## Eyeblink data analysis

For MF-ChR2 experiments videos from each trial were analyzed offline with custom-written MATLAB software (MathWorks). Distance between eyelids was calculated frame-by-frame by thresholding the grayscale image of the eye and extracting the count of pixels that constitute the minor axis of the elliptical shape that delineates the eye. Eyelid traces were normalized for each session, ranging from 1 (full blink) to 0 (eye fully open). Trials were classified as CRs if the eyelid closure reached at least 0.1 (in normalized pixel values) and occurred between 100 ms after CS onset and US onset. The average running for each animal was calculated by summing the average speed of each session (total distance run divided by session duration) and dividing by the total number of learning sessions, usually 20. Running speed for trial was calculated by dividing the distance run in the ITI preceding the current trial by the elapsed time. For all other experiments, MDMT data was processed using custom LabVIEW software (Neurasmus). Data was subjected to several quality checks: (i) trials with an unstable baseline were removed (trials with values exceeding 5x SD during the baseline period) and (ii) sessions with a large variation in UR amplitude (>0.5 x CV) were excluded. Finally, sessions from which more than 75% of trials were removed were excluded to ensure accurate behavioral measurements. Similarly to the MF-ChR2 experiments, CRs were defined as eyelid closures during the last 200 ms of the CS-US interval that exceeded 10% of UR amplitude. UR amplitude was calculated as the average of full eyelid closures during US-only trials and used to normalize the eyelid movements. Camera data, including data from electrophysiology experiments, were analyzed using

custom-made MATLAB scripts. Acquired data were preprocessed by filtering with a Gaussian low-pass filter (50 Hz cutoff frequency) and trials with unstable baselines (values > 5 x SD of the baseline period) were excluded. CRs were defined as eyelid closures during the last 200 ms of the CS-US interval exceeding 10% of the UR amplitude. In some cases, CRs were defined as eyelid closures exceeding values > 5 x SD of baseline values during the last 200 ms of the CS-US interval. CR percentage was calculated by dividing the number of CR trials by the total amount of eligible trials, multiplied by 100%. CR amplitude was calculated by dividing the maximum eyelid closure during the CS-US interval by the average UR amplitude, multiplied by 100%. CR amplitude at US onset represents the amplitude of eyelid closure at US onset (250 ms after CS onset. CR or EO onsets were manually detected on a trial-by-trial basis in a blind manner and represent the first data point before a continuously rising (CR) or decreasing (EO) signal.

## Spontaneous eyelid movement analysis

Spontaneous eyelid movements were identified during periods of recordings after removing the 2 s windows surrounding eyeblink trials. Eyelid movements were detected in Gaussian filtered (5 Hz cut-off) eyelid traces, as values of the first derivative that exceeded 3 x SD of the filtered trace. The 'findpeaks' MATLAB function with a minimum peak distance of 500 ms and minimum peak prominence of $1 \times 10^{-5}$ (arb. units) returned the positions of significant eyelid movements (openings or closures). Only isolated movements without openings or closures within a 500 ms window before the maximum velocity of the movement were included. Spontaneous eyelid onset was determined to be the first data point exceeding 2 x SD of the second eyelid derivative of the trace. Onset of $V_m$ depolarizations or hyperpolarizations were manually determined and were defined as the time point of the last minimum before a continuously increasing signal or decreasing signal, respectively. $V_m$ responses were tested for significance by ROC analysis on amplitudes surrounding the maximum eyelid velocity and preceding baselines.

## Microelectrode stimulation and Evans Blue labeling

Microelectrode stimulation was performed to determine the recording location targeting the eyeblink-encoding region of the CN. Eyelid movements were induced using an 80-μm-diameter platinum iridium monopolar electrode (100 KΩ; Alpha Omega), lowered into the brain with a 43° angle until a depth of 1500–1600 μm was reached. Electrodes were advanced in steps of 100 μm while currents were generated (ISO-flex Stimulus Isolater, A.M.P.I.) in the range of 1–15 μA (200 ms pulse trains; 250 μs biphasic pulses; 500 Hz). The depth at which reliable eyelid contractions could be observed with low current amplitudes (<10 μA) was identified as the eyeblink-encoding region of the CN. To mark this area in the CN, the electrode was retracted, coated with Evans Blue solution (5% in saline, Sigma-Aldrich) and reinserted using identical position and coordinates. Mice were transcardially perfused using 4% paraformaldehyde (PFA, Sigma-Aldrich) in phosphate buffered saline (PBS) after a lethal dose of pentobarbital (Nembutal) and brains were collected for histology.

## In vivo electrophysiology

Whole-cell and juxtasomal recordings were made using glass electrodes with a tip diameter of 1–2 μm and a resistance of 4–8 MΩ. Electrodes were heat-pulled on a P-1000 micropipette puller (Sutter Instrument) from filamented borosilicate glass capillaries (1.5 mm OD, 0.86 mm ID, Harvard Apparatus). Electrodes were back-filled with intracellular solution, containing (in mM): 10 KOH, 3.48 MgCl₂, 4 NaCl, 129 K-Gluconate, 10 HEPES, 17.5 glucose 4 Na₂ATP, and 0.4 Na₃GTP (295 - 305 mOsm; pH 7.2). The intracellular solution was supplemented with 0.5% Neurobiotin (Vector Labs). Electrophysiological recordings were made using a Multiclamp 700B amplifier (Axon Instruments, Molecular Devices) and digitized at 100 or 50 kHz with a Digidata 1440

digitizer (Axon Instruments). Electrodes were mounted with a 43° angle on a pipette holder connected to a head stage (CV-7B, Axon Instruments). Pipette movements were controlled by a micro-manipulator (SM7, Luigs und Neumann, Ratingen, GE). Pipettes were targeted at the AIP region of the CN and entered the surface of the cerebellum 1.5–2.0 mm lateral and -1.5 mm below Bregma, as previously described[50]. Electrodes were advanced while maintaining high pressure to a depth of 1500–1600 μm relative to the brain surface, after which the pressure was lowered to 15–20 mbar.

During whole-cell recordings, electrodes were advanced in steps of 2 μm while monitoring the current during 10 mV steps at 33 Hz in voltage-clamp mode without holding potential. All recorded neurons reached a seal resistance of at least 1 GΩ before establishing whole-cell configuration. Several protocols were run shortly after break-in to characterize the neuron, including a seal test protocol (10 mV steps at -65 mV) and a current step protocol (200 to -200 pA) in current clamp mode after compensation. Following these protocols, continuous recordings were made without current injection in current clamp mode while CS, US and CS-US paired trials were given (naive: 1:1:1 ratio in a randomized order, conditioned: repetitive sequences of 1US-5paired-1CS to maintain conditioned behavior throughout the experiment). Trial ITI was 12 ± 2 s and no threshold for eyelid opening was used to maximize the number of trials. Camera recordings of the eyelid were made simultaneously and synchronized with electrophysiology by recording TTL pulses controlling the CS and the US. Recordings were made until the neuron was lost or the recording quality became insufficient. Only one recording session was performed following the craniotomy surgery with a duration less than 4 h. During juxtasomal recordings, electrodes were advanced until action potentials ('spikes'; >1 mV) of a single neuron could be identified. Then, CS, US and CS-US trials were given while spiking responses were recorded. Juxtasomal recordings were made once per day, for 1–3 sessions each with a duration of less than 4 h.

## Calculation of electrophysiological parameters

Data were processed in Clampfit (v10.5, Axon Instruments) and MATLAB (R2011b and R2022a, MathWorks). Series resistance ($R_s$) and input resistance ($R_i$) were calculated during the seal test in voltage-clamp (holding of -70 mV), based on the peak current and steady-state current during a 10 mV voltage step, respectively. Membrane resistance ($R_m$) represents $R_i$ subtracted with $R_s$. Membrane time constant (tau) was estimated by fitting a single exponential function on the decay phase of the seal test or calculating the time between 100-37% of peak-to-steady state amplitude of the 10 mV voltage step. Membrane capacitance was calculated based on tau, $R_s$ and $R_m$. Resting membrane potential ($V_m$ rest) was estimated based on $R_s$ and the current necessary to maintain a holding potential of -70 mV in voltage-clamp. Spikes in $V_m$ recordings were detected and analyzed using Clampfit. Spikes were detected during threshold search on the first trace in the current step protocol that induced spikes. For some neurons we were unable to analyze spikes, because our current step protocol did not induce spikes. Since we prioritized $V_m$ measurements during EBC and spike rates were recorded juxtasomally in other recordings, we did not further drive the neuron to spike threshold.

## Exclusion and quality criteria in vivo whole-cell recordings

Recordings were discarded if $R_s$ exceeded 100 MΩ. Only traces that had a limited drift in $V_m$ were analyzed. Some recordings were affected by an incorrect $V_m$ offset directly visible when switching to current-clamp, caused by an issue with the reference electrode. Since we analyzed $V_m$ responses relative to baseline, we included these neurons. Most recordings were without current injection, but a few neurons received limited current (−25 to −200 pA) to maintain $V_m$ at original values directly after break-in. Average recording depth of all neurons (1812.4 ± 33.4 μm) was consistent with the depth of the CN[50]. Neurons

were recorded in both hemispheres (Supplementary Table 1) and we did not find a difference in CS-evoked $V_m$ amplitude between sides for either group (cond: $P = 0.77$, naive: $P = 0.571$, pooled: 0.7, unpaired $t$ test).

### Spike analysis of in vivo juxtasomal recordings

Spikes were detected using a custom-written MATLAB spike analysis program (B.H.J. Winkelman, Netherlands Institute for Neuroscience, Amsterdam), after which spikes were sorted based on spike waveform characteristics. Only data with spikes from single units were included in the analysis. Based on spike times we calculated: spiking frequency ( = number of spikes / duration of time period) and coefficient of variation of spiking (CV = standard deviation (all inter-spike-intervals (ISIs) / average (all ISIs)). Spontaneous spike calculations were based on time periods between trials.

### Processing of $V_m$ and eyelid data

Electrophysiology and eyelid data were loaded into MATLAB using the '*abf2load*' (H. Hentschke, University of Tuebingen and F. Collman, Princeton) and the '*TDTbin2mat*' function, respectively, and further analyzed using custom MATLAB code. In short, TTL timings were used to synchronize both types of data. In $V_m$ traces spikes were removed by median filtering (10 ms window). Trial epochs were normalized to baseline (250-0 ms) before the CS. CS-evoked responses were obtained by pooling paired and CS-only trials, and CS-responses were analyzed until US onset (250 ms after CS onset). No separate analysis of CS-only trials was performed because of the low number compared to the number of CS-US paired trials. To establish whether a neuron showed responses to a stimulus we used non-parametric ROC analysis on trial-by-trial amplitudes during the baseline (250-0 ms before CS onset) vs. the period of interest (CS: 50–250 ms after CS; US: 0–500 ms after US). The overlap between amplitude distributions was quantified as area under the ROC curve (AUC). Bootstrap analysis (1000 repeats) with shuffled data from those distributions was used to test for statistical significance between AUC distributions. Amplitude and peak time of the CS-evoked $V_m$ response was determined on a trial-by-trial basis 50–250 ms after CS onset. Area was calculated as the sum of values over time periods indicated in the text. Amplitude, peak time and area of the US-evoked $V_m$ response was determined on a trial-by-trial basis between 0–500 ms after US onset. Data per neuron were obtained by averaging trial quantifications. Onset of $V_m$ depolarizations or hyperpolarizations were manually determined and were defined as the time point of the last minimum before a continuously increasing signal or decreasing signal, respectively. Onsets of CRs or EOs were manually determined in a similar way, while the experimenter was blind to the coinciding $V_m$ trace. To classify neurons as CR-dependent or CR-independent, we performed ROC analysis on the above time periods, comparing CR vs. NR trials. For a few neurons there were insufficient CR or NR trials to be able to perform ROC analysis and in these neurons we instead considered the relationship between $V_m$ amplitude and CR amplitude, in order to determine whether the occurrence of CRs coincided with increased $V_m$ responses.

### Whole-body video recordings

Videos were acquired using iPi Recorder 4 software at 60 fps (800×600 pixels) using an infrared-integrated camera (ELP-USBFHD05MT-KL36IR, ELP). Videos were converted to AVI format using iPi Mocap Studio 4 and analyzed in MATLAB. The sum value of the first derivative between two subsequent frames was calculated as a measure of movement. The trace was filtered using a Gaussian filter (25 Hz cut-off) and normalized using the median and the max values across the recording. CS onsets were identified by reflection of the LED on the wall of the box. Amplitude and area were calculated between 0–250 ms after CS onset (CS quantifications) and 0–500 ms after US onset (US quantifications).

### Optogenetics

During MF-ChR2 experiments, optogenetic channels were stimulated using 473 nm laser light (LRS-0473-PFF-00800-03, Laserglow Technologies), controlled by custom-written LabVIEW (National Instruments) software. Laser power was adjusted for each mouse and controlled for each experiment using a power meter (Thorlabs) at the beginning and end of each session. With these stimulation parameters, stimulation is likely to be highly localized (predicted spread from cortex to AIP sites of <1% based on https://nicneuro.net/optogenetics-experimental-tools/). For activation of ChR2 in mossy fibers as a CS, laser power was lowered below the threshold for detectable eyelid movement upon repeated stimulus presentation. The optogenetic CS consisted of stimulation that lasted for 350 ms (or 550 ms in the timing experiments) delivered in trains of 2 ms pulses and was paired with a co-terminating 50 ms air-puff US. Laser power ranged from 0.20 to 1 mW, corresponding to intensities (power output per unit area) of 6 to 30 mW/mm2 for stimulation within the cerebellar cortex, and of 3 to 20 mW/mm2 for stimulation in the AIP.

In experiments using optogenetic stimulation of PN neurons 465 nm light-induced activation of Chronos was achieved using a 5 W LED (60 lm, LZ1-B200, LED Engin Inc) connected to a custom-made LED driver[78]. This resulted in a light output of 1–2.5 mW at the tip of the optic fiber implants. LEDs were controlled by TTL pulses from the amplifier and these pulses were used to synchronize electrophysiology and camera recordings. Silver conductive paint and aluminum foil coating was used to reduce the escape of optogenetic light from the fibers and equipment. The AUC for the optogenetically-evoked eyelid responses was calculated as the absolute sum of the normalized-to-baseline eyelid trace 0–150 ms after LED onset. Optogenetic stimulation of ChR2(H134R) in Gabra6-Cre::ChR2EYFP mice was achieved using a 465 nm 5 W LED (60 lm, LZ1-B200, LED Engin Inc) mounted on a moveable arm. After establishing a whole-cell recording, the LED was positioned close to the cerebellum to activate granule cells.

### Histological procedures

To examine fiber placement after MF-ChR2 experiments and to verify hM4Di-DREADD expression, mice were deeply anesthetized using a ketamine-xylazine mixture and perfused transcardially with 4% paraformaldehyde and their brains were removed. Sagittal or coronal sections (50 μm thick) were cut using a vibratome. Slices from the DREADD transgenic mice were stained with a rabbit monoclonal anti-HA antibody (1:400, Cell Signaling, #3724) and DAPI (abcam, #ab104139). Slices were then mounted on glass slides with Mowiol mounting medium (Merck Millipore Calbiochem). For all other histological quantifications, mice were deeply anesthetized with pentobarbital (Nembutal) and transcardially perfused with 100 ml 4% paraformaldehyde (PFA) in 0.12 M phosphate buffer. Brains were dissected and post-fixed overnight at 4 °C, then cryoprotected in 0.12 M phosphate buffer containing 30% sucrose at 4 °C until they sank. Cerebella were cut on a cryostat into 25 μm-thick coronal or sagittal floating sections and collected in phosphate-buffered saline (PBS). Primary antibodies were incubated overnight at 4 °C in PBS containing 0.25% Triton X-100, and 5% fetal calf serum. Primary antibodies were: mouse anti-NeuN (1:500, Millipore, # MAB377), rabbit anti-GFP, which also detects YFP (1:1000, Chemicon, # AB3080), chicken anti-GFP (1:1000, Aves labs, GFP-1020), mouse anti-calbindin (1:1500, Swant, # 300), guinea-pig anti-VGLUT1 (1:500, Synaptic Systems, # 135304), rabbit anti-VGLUT1 (1:1000, Synaptic Systems, # 135302), guinea pig anti-VGLUT2 (1:1000, Synaptic Systems, # 135404) and mouse anti-gephyrin (1:500, Synaptic Systems, # 147021). After washing, sections were then incubated for 1 h at room temperature with one of the following fluorophore-conjugated secondary antibodies or streptavidin (diluted 1:1000): donkey anti-mouse Cy3 (Jackson Immunoresearch, # 715-165-150), goat anti-chicken Alexa Fluor 488 (ThermoFisher Scientific, # A-11039), donkey anti-guinea pig Cy3 (Jackson Immunoresearch,

# 706-165-148), donkey anti-rabbit Alexa Fluor 647 (ThermoFisher

# 706-165-148), donkey anti-rabbit Alexa Fluor 647 (ThermoFisher Scientific, # A-31573), donkey anti-rabbit Alexa Fluor 488 (Thermo-Fisher Scientific, # A-32790), streptavidin Alexa Fluor 488 (Thermo-Fisher Scientific, # S11223) or streptavidin Cy3 (Jackson Immunoresearch, # 016-160-084). For each immunohistochemical reaction, slices from all experimental conditions were processed together and incubation times were kept constant. After processing sections were mounted on microscope slides with Tris-glycerol supplemented with 10% Mowiol (Merck Millipore Calbiochem).

## Imaging
Fluorescent images were acquired using a confocal microscope (SP5 and SP8, Leica Microsystems) or an upright Zeiss Axio Imager M2 microscope using a 10x objective. Confocal images were taken at a resolution of 1024 × 1024 dpi and a 50 Hz speed. Laser intensity, gain and offset were maintained constant in each analysis. Quantitative evaluations were made by a blind experimenter using Fiji software. Adobe Photoshop 6.0 (Adobe Systems) was used to adjust image contrast and assemble the final plates.

## Quantification of density of glutamatergic terminals
To estimate the density of VGLUT1 and VGLUT2-positive axon terminals, at least three coronal sections containing the AIP (Bregma: −6.00, −6.12 mm) were selected for each animal. In each section three 1 μm-thick confocal images/side were captured under a 63x objective. In Fiji, a "maximum intensity" function was applied to the Z-projection and the "analyze particle" function was used to estimate the density of boutons (number of terminals/mm$^2$), after selecting the automatic threshold and using the "watershed" function. Only particles with a size between 0.4 μm$^2$ and 3 μm$^2$ were included in the analysis. The average/animal in the ipsilateral and contralateral side was calculated (VGLUT1: conditioned mice, $N = 11$; pseudo-conditioned mice, $N = 13$; VGLUT2: conditioned mice, $N = 4$; pseudo-conditioned mice, $N = 6$). Histological procedures were performed at the same time for the pseudo-conditioned and conditioned mice within each batch of mice. To correct for unwanted differences between batches of mice that were trained at different time points, values were normalized to the values of the pseudo-conditioned group for each batch of mice.

## Quantification of density of axonal varicosities
The density of axonal varicosities in the AIP was evaluated in the images used for VGLUT1 quantifications (see paragraph above) where YFP+ axons could be detected by measuring in each image the length of individual YFP+ axonal segments and the number of enlargements on each segment (conditioned mice: $N = 3$, $n = 322$ axonal segments; pseudo-conditioned mice: $N = 3$, $n = 381$ axonal segments). The density for each side was calculated by dividing the total number of varicosities by the total length.

## Quantification of gephyrin-positive puncta
Single 0.4 μm thick-confocal images of the AIP (ipsilateral and contralateral side) in coronal sections (Bregma: -6.00, -6.12 mm) were collected under a 63x objective with 2.5x zoom. At least three sections were selected/mouse. On such images the density of gephyrin-positive puncta around the soma of excitatory neurons (identified by their size, i.e. > 240 μm[43]) was evaluated by Fiji and expressed as number/μm neuronal membrane. Gephyrin-positive puncta were manually identified, as they were clearly visible as discreet fluorescent puncta. At least 15 neurons/mouse were quantified (conditioned mice: $N = 4$; $n = 84$; pseudo-conditioned mice: $N = 6$, $n = 110$). For each neuron the number of gephyrin-positive puncta apposed to calbindin-positive terminals was quantified and the % of those puncta (out of the total number of puncta) was calculated. As mentioned above, to correct for unwanted differences in different batches of mice, gephyrin densities were normalized to the values of the pseudo-conditioned group in each batch of mice.

## Quantification of size of rosettes
Pictures including YFP+ rosettes in the eyeblink area (i.e. at the bottom of the primary fissure) consisted of thirteen 0.5 um-thick confocal steps acquired under a 63x objective. Three pictures were taken to cover the eyeblink area in each side. The "maximum intensity" function in Fiji was applied to the Z-projection and the area of individual rosettes was quantified (conditioned mice: $N = 5$, $n = 613$; pseudo-conditioned mice: $N = 4$, $n = 761$).

## Decoding analysis
Linear discriminant analysis (tenfold cross-validation) was performed on pooled trial data with a varying number of trials for each neuron (cond: $n = 199$ CR and $n = 265$ NR trials in 15 neurons; naive: $n = 113$ EO and $n = 214$ NR trials in 15 neurons). For whole-cell data decoding was performed on spike-removed, normalized and Gaussian filtered (10 ms cut-off) $V_m$ traces. For spike data, spikes were binned into 5 ms time intervals, Gaussian filtered (10 ms cut-off) and normalized to baseline rate. The dataset was then divided in a train and test set (9:1 ratio) and decoding was performed using the 'classify' function in MATLAB (modified from MATLAB code kindly provided by M. Tang, Australian National University, 2022). Accuracy represented the percentage of accurately classified trials for each sampling point or time bin. To test whether the decoder performed above chance level, we performed a cluster-corrected permutation test with 2000 permutations in MATLAB (E. Spaak, Oxford University, 2015). To calculate decoding accuracy for decreasing group sizes, we performed 29 iterations of the decoding analysis where in each iteration a random subsample of the total dataset was selected using the 'randsample' function in MATLAB. Quantifications and traces for each group size were calculated by averaging across 29 iterations. Time period in which above-average decoding took place was calculated as the average time the cluster-test was significant.

## Statistics
Statistical analysis was carried out using the Statistics toolbox of MATLAB (R2011b and R2022a, MathWorks, CA, USA), SPSS (v22, IBM, NY, USA), GraphPad Prism 5 and 9 (GraphPad Software Inc., CA, USA). Normality of distributions was assessed using Shapiro-Wilk test. Homogeneity of variance was assessed using Bartlett's test. If the assumption of normality was violated, non-parametric alternatives were performed. For the correlation between walking speed vs. onset session we used linear regression analysis and a unpaired $t$ test. To compare average peak timings for the different ISIs we used a unpaired $t$ test. For MF-ChR2 experiments, mice were randomly assigned to specific experimental groups without bias and no animals were excluded. Behavioral performance was analyzed using a repeated-measures ANOVA or linear mixed models with the maximum likelihood method. Group and session/day was modeled as fixed effects, and %CR, CR amplitude and CR amplitude at US onset were modeled as dependent variables. We assessed the fit of the model by running the analysis with the unstructured, diagonal and first-order autoregressive repeated covariance types, after which we choose the covariance type with the lowest Akaike's information criterion (AIC) value. Correlations between eyelid and $V_m$ or spike data were performed using Spearman correlation, where data were normalized to the maximum value per neurons. Differences in correlations were tested using the Fisher's Z-transformation (J. Decoster & A-M. Leistico, University of Virginia, 2005). Distributions of CS responses between conditioned and naive mice were tested using the chi-square test. Histological data were analyzed using a paired $t$ test (when comparing ipsilateral and contralateral sides), unpaired $t$ test (when comparing conditioned with pseudo-conditioned mice), Pearson correlation analysis and

# Article

Kolmogorov-Smirnov test (when comparing cumulative frequencies). Data are reported as mean ± standard error of the mean (SEM), unless indicated otherwise. Statistical power was calculated using Gpower (version 3.1.9.4). Data are shown in figures as mean ± SEM., unless indicated otherwise. Differences were considered significant at *$P < 0.05$, **$P < 0.01$ and ***$P < 0.001$.

## Reporting summary

Further information on research design is available in the Nature Portfolio Reporting Summary linked to this article.

## Data availability

Data files are available from the corresponding authors upon request. Source data are provided with this paper.

## Code availability

The code used for the analysis of our experiments are available at https://github.com/BroRobin/SynMechCN.

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

## Acknowledgements

We are grateful to the following people for their contribution to the study: J.W. Potters, H.J. Boele, S.K.E. Koekkoek, Z. Gao, M.M Ten Brinke, J.K. Spanke, A.C.H.G. IJpelaar, M. Tang, B. Winkelman, A. Court, C. Geelen, M. Mešković, G. Özel, M. Bouw, S.O. Stokman, M. Romijn, S. Dijkhuizen and T. Jacobs. We thank Tracy Pritchett and Ana Machado for maintenance of mouse lines. Catarina Carvalhas assisted with some of the data acquisition. We thank Jovin Jacobs for helpful discussions throughout the project. We thank I. Duguid and P. Chadderton for their constructive comments. This study was enabled by funding from the Netherlands Organization for Scientific Research (NWO-ALW 824.02.001; C.I.D.Z., NWO STEM - VBT 2021 19224 and NWO 863.14.005; C.B.C.), the Dutch Organization for Medical Sciences (ZonMW 91120067; C.I.D.Z.), Medical Neuro-Delta (MD 01092019–31082023; C.I.D.Z.), INTENSE LSH-NWO (TTW/00798883; C.I.D.Z.), ERC-adv (GA-294775 C.I.D.Z.) and ERC-POC (nrs. 737619 and 768914; C.I.D.Z.), the NIN Vriendenfonds for Albinism (C.I.D.Z.), the Dutch NWO Gravitation Program, Dutch Brain Interface Initiative (DBI2; C.I.D.Z.), the Boehringer Ingelheim Fonds, a Howard Hughes Medical Institute International Early Career Scientist Grant #55007413 (M.R.C.), European Research Council Starting (#640093) and Consolidator (866237) Grants (M.R.C.) and a fellowship from the Portuguese Fundação para a Ciência e a Tecnologia SFRH/BD/77686/2011 (C.A.).

## Author contributions

R.B., C.A., C.B.C., M.R.C., C.I.D.Z. conceived the project and designed the experiments. R.B. performed and analyzed the awake and anaesthetized whole-cell and juxtasomal recordings, pontine nuclei optogenetics, behavioral trainings for histological investigation, whole-body video recordings and decoding analysis. C.A. performed and analyzed the optogenetic mossy fiber ChR2 and chemogenetic experiments. D.C. performed and analyzed the histological procedures. R.B., C.A., D.C., C.B.C. M.R.C., C.I.D.Z. wrote and edited the manuscript. C.B.C., M.R.C., C.I.D.Z. provided supervision. C.I.D.Z., C.B.C. and M.R.C. acquired the funding.

## Competing interests

The authors declare no competing interests.

## Additional information

**Peer review information** : *Nature Communications* thanks Dimitar Kostadinov, Dieter Jaeger and the other, anonymous, reviewer(s) for their contribution to the peer review of this work. A peer review file is available.

