## [Peer Review File · Nature Communications]

Synaptic mechanisms for associative learning in the cerebellar nucleiReviewers' comments:

Reviewer #1 (Remarks to the Author):

Summary:

In this manuscript, Broersen and colleagues study the changes occurring in cerebellar nuclear (CN) neurons as mice undergo Pavlovian eyeblink conditioning. To address their question, the authors utilize an impressive combination of techniques: combining whole-cell patch clamp and extracellular electrophysiology during behaviour, histological quantification of synaptic proteins, and optogenetics. The experiments, in particular the whole-cell recordings in behaving mice, are highly challenging and generally well-executed. The authors draw the following conclusions from their work:

(1) As mice begin to blink predictively in response to the conditioned stimulus, CN neurons gain a (usually excitatory) movement-predictive signal that can be used to decode whether a mouse exhibited a conditioned response on a given trial,

(2) This functional change correlates with structural changes in the cerebellar nuclei.

Notably, the density of excitatory synaptic puncta from the mossy fiber pathway increases (although only on the contralateral side).

(3) With learning, the potency with which pontine (mossy fiber) stimulation can directly drive eye blinks increases, supporting the idea that this pathway is facilitated during learning.

Overall, this study provides an important addition to our understanding of how cerebellar activity is modified during eyeblink conditioning and how neural circuits may be altered by learning in general. My main concerns are regarding the not quite solidified relationship between the functional changes described in Figures 1, 2, and 5 and the structural plasticity measurements in Figure 4. Despite these concerns, I am generally enthusiastic about the work and recommend the authors are given the opportunity to revise their manuscript.

Major points:

1. A strength of this study is its ability to connect functional changes in the cerebellar nuclei at the level of synaptic currents and spiking to structural changes as measured by staining of synaptic puncta. However, the functional and anatomical findings relating to changes in

excitatory inputs in CN neurons are at odds with the anatomical findings that show structural plasticity on the contralateral side. The authors mention in the discussion that this result may be explained by the fact that eyeblink conditioning in mice leads to a whole-body defensive behaviour. However, this is not quantified anywhere in the study – only eye movements are measured. The authors should elaborate on this point, as it is arguably the key to unifying the findings of this study.

2. In Figure 4 – the authors first demonstrate an increase in density of gephyrin puncta formed on CN neurons. They then show that there is not a significant decrease in the density of gephyrin puncta that are not apposed by calbindin-positive presynaptic terminals. Together, they use this to argue that the increase in gephyrin puncta must be due to an increase in puncta that are apposed by calbindin-positive presynaptic terminals. Is there a reason why the authors did not simply measure the terminals in question (i.e. Purkinje cell inputs) directly?

3. Related to point 2, it is not clear how appositions were quantified and what threshold was used to determine if gephyrin puncta faced or did not face a calbindin-positive terminal. Please explain and justify this analysis.

4. In Figure 4H-I, the authors show that spontaneous Purkinje cell firing rates are higher in conditioned animals than in naïve ones. They interpret this in the context of the structural changes they observe in ponto-nuclear projections (Paragraph starting on line 292). This seems like a strong overinterpretation of their results, as no direct measurements of structural changes in the cerebellar cortex have been made. Moreover, it is not obvious (to me) that an increase in mossy fiber input would necessarily lead to higher rates of Purkinje cell spiking. The cerebellar cortex goes to great lengths to balance excitation and inhibition, so it is equally plausible that this 'extra' mossy fiber input is cancelled by cortical interneurons before it reaches Purkinje cells. I would suggest tempering this claim or demonstrating directly that mossy fibers drive stronger Purkinje cell spiking after learning.

5. In Figure 5, the authors demonstrate nicely that pons stimulation can evoke eyelid closures after training but has essentially no effect pre-training. This interesting result begs

several questions:

Related to point 1, do the authors observe whole-body defensive movements when they stimulate the pons after training (as in Figure 5)? Would the results of this experiment look different if the photostimulation was done directly in the DLH or IntA regions of the cerebellar nuclei?

Does pons stimulation in trained mice result in stronger spiking responses in CN neurons?

Minor points:

1. Abstract – This sentence is not grammatically sounds: ‘Here we studied these changes in inputs using whole-cell and juxtosomal recordings of cerebellar nucleus neurons in awake mice during eyeblink conditioning and doing post-learning structural analysis.’ I think the easiest fix is to add the word ‘by’ between ‘inputs’ and ‘using’.

2. ‘EPSP/IPSP neuron’ nomenclature – it seems presumptuous to use these names to refer to the neurons activated and suppressed by the CS. As the authors themselves point out, there could be different mechanisms that lead to the membrane dynamics they observe (e.g. disinhibition – line 94). I would suggest the authors switch their naming terms to something more descriptive like CS-activated and CS-suppressed neurons.

3. Line 285 – technically, ‘VGLUT1 expression’ (i.e. mRNA) was not measured here. Please change to a more accurate term like ‘VGLUT1 puncta density’

4. Figure 4 and other places – it would be more intuitive if the authors referred to the left and right hemispheres as ipsilateral and contralateral to the side of the conditioning. This is the more relevant comparison that left and right, and exhibiting the data in this way does not require a reader to remember the side on which the animals were conditioned.

5. Figure 5F – It looks like this is an average across animals (according to the legend) but there are no error bars on these traces. Please add these.

6. Line 735 – coordinates should be in millimeters, not micrometers.

Reviewer #2 (Remarks to the Author):

In this manuscript Broersen et al utilize the technique of whole cell and cell attached single cell recording from the cerebellar nuclei of mice while they perform a conditioned eyeblink behavior. Additionally, the authors use antibodies to examine for changes in excitatory and inhibitory connections in the CN after conditioning, and they use optogenetic activation of pontine nuclei to determine changes in mossy fiber input after conditioning.

The most significant insight we do get, albeit with very small N's and on somewhat shaky statistical grounds, is that depolarization in DHP and/or IntA neurons during the CS in conditioned animals precedes spike rate changes and is enhanced after conditioning. In addition, a linear decoder shows that this early activity has modest predictive information on the ensuing behavior.

Major Comments

1. In vivo whole cell recording from deep structures in behaving mice is an incredibly difficult technique, which explains the really low sample number of neurons obtained (6 from naïve mice, and 8 from conditioned mice). The authors do themselves a bit of a disservice by then being very lax about rigor in their statistical analyses of these numbers. It is never mentioned how many mice these neurons are from, which is a critical piece of information. Nor do they show which ones are from the DLH or IntA nucleus, which is important in light of differential antibody staining findings between the 2 nuclei. A number of statements about changes seen are not supported by statistical significance, which may be unsurprising given the small N's. Scientific rigor would require that for each results statement, the n of neurons, the n of mice, and the nuclei involved would be stated.

2. One of the major advantages of whole cell recordings is that one could discern excitatory from inhibitory postsynaptic responses by injecting a bias current for some of the traces, or perform a voltage clamp analysis. Such methods have for example been utilized by Destexhe et. al. The authors do not perform this test and instead call any depolarizing Vm deflection an EPSP, although this could either reflect disinhibition by reduced PC input or increased

excitation by MF input. While this is acknowledged by the authors in some remarks, it is not following the usual definition of an EPSP, which should be due to excitatory input, not disinhibition. As a consequence, the answer that would be most interesting to get, namely if MF excitation to CN during the CS in response to conditioning is responsible for the added depolarization seen, is not given.

3. The optogenetic stimulation of the pontine nuclei also affects both input to the cerebellar cortex and the cerebellar nuclei, not allowing any disambiguation of the relative pathway significance in this behavior. Optogenetic inhibition of mossy fiber terminals from the PN in the CN would have been a much stronger experiment to support the functional role of excitatory input to these neurons in conditioned depolarizations.

Detailed comments:

1. Table 1 should include columns, for animal ID, and nucleus recorded.
2. Line 77/78. Example of results statement without significance due to low n.
3. Line 96/97. I don't understand how CS responses can be later in CS only trials than in CS/US trials, given that the mice have no information on whether a US will follow, or do they? The Methods are not clear on the sequencing of trial types during recording sessions (lines 581-594).
4. Line 102. Example of missing statement of N's. "For all neurons for which we recorded both CR+ and Cr- trials ..."
5. Line 104/106. Reporting results from a single neuron becomes really underpowered.
6. Line 119/120. Another effect statement without N's. (numbers of mice, numbers of neurons, which nucleus?)
7. Line 191/192. With a p-value of 0.031 we learn that the contralateral hemisphere has increased VGLUT1+ puncta compared to the ipsilateral (which carries the conditioned eyeblink response). This counterintuitive result may be spurious given the small N's (in this case of mice processed with antibodies). How many statistical tests were made in total, and may this outcome be a result of p-fishing? A Bonferroni or similar adjustment should be made for the number of total tests attempted.

Reviewer #3 (Remarks to the Author):

This manuscript by Broersen and colleagues aims to examine structural and synaptic changes in the connection from mossy fibers to cerebellar nucleus (CN) neurons following eyeblink conditioning, a form of associative learning that is dependent on the cerebellum. The authors use a variety of techniques including in vivo whole cell patch clamp and juxtosomal recordings during awake behavior, immunohistochemistry, and optogenetics to look for changes in this pathway following learning. They find some evidence that is suggestive of learning-induced remodeling of mossy fiber--but not climbing fiber--inputs to CN, including increased EPSPs in response to the conditioned--but not unconditioned--stimulus, increased expression of a marker of mossy fiber--but not climbing fiber--terminals, and a progressive increase in eyeblink size driven by optogenetic stimulation of pontine mossy fibers during conditioning. The mossy fiber to CN pathway is an understudied and potentially important site of plasticity underlying cerebellar-dependent learning so the topic addressed by the paper should be of high interest to the cerebellar community. In addition, the attempt to link structural changes with synaptic properties examined electrophysiologically increases the potential impact of the paper. However, the study suffers from several conceptual and methodological issues that prevent the experiments from supporting the conclusions and detract from its overall significance to the field.

Major

* The number of neurons recorded by whole cell patch clamp does not seem sufficient to support the conclusions. I understand that these are very difficult experiments to perform, but the decision to record from both the ipsilateral and contralateral CN in conditioned mice would seem to have unnecessarily reduced statistical power given what we know about the *ipsilateral* control of movement by the cerebellum. If the goal was to look for structural and synaptic plasticity in the neurons that are responsible for driving the learned behavior it would have been better to focus on the ipsilateral (to the air puff) IntA and DLH, which are more established as being essential for generating learned eyeblinks (however, see related comment below regarding the histology results). Based on Fig. 1f there only appear to be about 2-3 neurons that have consistent EPSPs during the ISI in conditioned mice. These numbers are unlikely to allow any kind of meaningful statistical tests and render statements

about the relative changes in EPSPs for conditioned and unconditioned stimuli inconclusive. The pooling of trials does not alleviate this issue because it's not clear what proportion of trials is coming from each neuron and how this is accounted for in the statistics. The low "n" is especially problematic because the whole cell patch clamp experiments are a major part of the paper and arguably the biggest novelty.

* The structural changes revealed by VGLUT1/2 expression are hard to interpret based on what we know about the eyeblink-controlling regions of the CN. Given the importance of the ipsilateral IntA/DLH in driving the learned eyeblinks, it's hard to make the case that the changes in mossy fiber input to the contralateral CN are driving the learned responses. The authors cite Heiney et al, 2021 to say that movements of multiple body parts are learned during eyeblink conditioning but this paper actually shows that all of the movements are still driven by the ipsilateral IntA. It needn't be the case that the only plasticity relevant to the learned behavior is happening in the "eyeblink controlling" part of the CN but it does seem problematic if there is no evidence of plasticity there. In my mind, this substantially decreases the significance.

* There is also little attempt to relate the structural changes in mossy fiber innervation to the electrophysiology. Were the neurons with large EPSPs during the ISI recorded in the same region/side as the increased VGLUT1 was found? What about the juxtacellularly recorded neurons with increased spiking during the ISI? This is glossed over in the presentation of the results. The data would be easier to interpret as a whole if more effort was made to tie together the results from the various experiments in the paper.

* Related to these points, the learning in many of the conditioned mice appears to be poor (the insets in many figures show average CR amplitudes of 10% full closure, which is the minimum threshold the authors use to define a CR). If the mice didn't learn well how can we interpret any of the findings from the electrophysiology and histology experiments, especially given the small sample sizes and relatively large responses of the neurons in naive mice? The study seems generally underpowered for the conclusions.

* The juxtosomal recordings do not appear to add much to what is already known in the

literature. The main value of including these recordings would seem to be to relate them to the whole cell recordings but the connection between these two is underdeveloped. Were cell attached recordings ever made before going whole cell, which might allow a more direct cell-by-cell comparison of input-output? As is, there is a reliance on a population analysis of cells recorded using each method but since the n for the whole cell recordings is so small it's hard to appreciate the connection to the juxtosomal recordings.

* The decoder results aren't particularly enlightening given the small sample sizes because presumably a decoder would perform better if it were "listening" to more neurons than were included in the sample (especially for the whole cell recordings).

* The US coding results suffer from the same issue. It's hard to prove the negative when there is a hint of a difference between groups that could become significant if more cells were included. There is no power analysis provided to suggest that the sample size is appropriate to answer the question.

* There is no discussion of alternative plasticity mechanisms. The authors focus solely on (presynaptic) mossy fiber structural plasticity and attempt to link it to the observed CS-driven EPSPs in CN neurons but other potential plasticity mechanisms, including post-synaptic changes in CN neurons, are not considered or discussed.

* The optogenetics experiment as performed cannot answer the question it sets out to answer because stimulating in the PN will also activate mossy fibers projecting to the cerebellar cortex, which could lead to larger eyeblinks due to plasticity within the cortex regardless of changes in the CN. In addition, the experiment is missing a couple of critical controls, which makes it impossible to interpret. There needs to be a control for increased Chronos expression over time, which could also explain the results. A pseudoconditioned group would be one possibility but perhaps there are others as well. It also needs a GFP or similar control to make sure the mice aren't just blinking to the laser light as they learn (since the CS is also a light), although depending on the time window analyzed this may not be necessary.

Minor

* Given the small n for many of the experiments wouldn't it be more appropriate to report medians and ranges rather than means and SEMs for the summary statistics?

* When using a paired t-test with missing data from one group, wouldn't the "n" be the lesser n of the two groups, i.e. only those cells without missing data?

* Figure 1

e: It's hard to see the eyelid traces when the voltage traces are displayed on top of them.

i/j: It would help to show different neurons with different markers to appreciate the relative proportions.

k: On the printed page, it's hard to tell the difference between the markers for the "cond IPSP" and "naive" groups.

* Figure 4e/f: Wouldn't it be more direct to quantify Gephyrin+ puncta that ARE apposed to CALB+ boutons as "PC" terminals rather than the way it was done? Is there a reason why this can't be done? The reasoning seemed convoluted.

* There are several references that appear inappropriate/incorrect. For example, ref 23 on page 2 line 44, ref 33 on page 3 line 87, and ref 50 on page 7 line 220 are cited for things that they don't appear to have anything to do with, but there may be others as well.

* The logic of the following statement is not clear to me (page 7 line 218): "The increased density of VGLUT1+ MFs in the DLH after conditioning may suggest that conditioning is also facilitated by an overall increased MF drive to the cerebellar cortex." Can this be elaborated?

Reviewer #1

In this manuscript, Broersen and colleagues study the changes occurring in cerebellar nuclear (CN) neurons as mice undergo Pavlovian eyeblink conditioning. To address their question, the authors utilize an impressive combination of techniques: combining whole-cell patch clamp and extracellular electrophysiology during behaviour, histological quantification of synaptic proteins, and optogenetics. The experiments, in particular the whole-cell recordings in behaving mice, are highly challenging and generally well-executed. The authors draw the following conclusions from their work:

- (1) As mice begin to blink predictively in response to the conditioned stimulus, CN neurons gain a (usually excitatory) movement-predictive signal that can be used to decode whether a mouse exhibited a conditioned response on a given trial,
- (2) This functional change correlates with structural changes in the cerebellar nuclei. Notably, the density of excitatory synaptic puncta from the mossy fiber pathway increases (although only on the contralateral side).
- (3) With learning, the potency with which pontine (mossy fiber) stimulation can directly drive eye blinks increases, supporting the idea that this pathway is facilitated during learning.

Overall, this study provides an important addition to our understanding of how cerebellar activity is modified during eyeblink conditioning and how neural circuits may be altered by learning in general. My main concerns are regarding the not quite solidified relationship between the functional changes described in Figures 1, 2 and 5 and the structural plasticity measurements in Figure 4. Despite these concerns, I am generally enthusiastic about the work and recommend the authors are given the opportunity to revise their manuscript.

We thank the Reviewer for the positive and constructive feedback.

- A strength of this study is its ability to connect functional changes in the cerebellar nuclei at the level of synaptic currents and spiking to structural changes as measured by staining of synaptic puncta. However, the functional and anatomical findings relating to changes in excitatory inputs in CN neurons are at odds with the anatomical findings that show structural plasticity on the contralateral side.

We have addressed this issue in two ways. First, we have trained new mice (a conditioning and a pseudo-conditioning group) and quantified for synaptic biomarkers. In this elaborated study we find that VGLUT1+ excitatory puncta in the CN are increased on both sides without a significant difference between them. Second, to isolate VGLUT1+ MFs specifically, we have made fluorescent tracer injections in the pons bilaterally. Here we found that YFP+ varicosities are also increased bilaterally, further supporting our VGLUT1+ findings. Now all data sets are compatible.

- The authors mention in the discussion that this result may be explained by the fact that eyeblink conditioning in mice leads to a whole-body defensive behaviour. However, this is not quantified anywhere in the study – only eye movements are measured. The authors should elaborate on this point, as it is arguably the key to unifying the findings of this study.

We agree with the reviewer, as it directly relates to recent findings on a putative role of anterior interpositus nucleus neurons in coordinating multiple body parts (Heiney et al. 2021). To address this comment we have performed additional conditioning and pseudo-conditioning experiments while acquiring video recordings. Our analysis shows that conditioned mice indeed develop a whole-body response to the CS over the course of training and that these movements correlate with the eyelid response in the majority of the mice (Supplementary Fig. 7). Importantly, this is not observed in pseudo-conditioned mice. These data have now been added to the revised manuscript.

- In Figure 4 – the authors first demonstrate an increase in density of gephyrin puncta formed on CN neurons. They then show that there is not a significant decrease in the density of gephyrin puncta that are not apposed by calbindin-positive presynaptic terminals. Together, they use this to argue that the increase in gephyrin puncta must be due to an increase in puncta that are apposed by calbindin-positive presynaptic terminals. Is there a reason why the authors did not simply measure the terminals in question (i.e., Purkinje cell inputs) directly?

*We thank the Reviewer for bringing this up. We quantified gephyrin puncta not apposed by CALB-positive terminals, as they only make up 2-3% of the total number of gephyrin puncta. These puncta were much easier to count than those that are apposed by CALB-positive terminals because of their lower numbers. We reasoned that there are two types of gephyrin+ inhibitory terminals, those that form the connection with Purkinje cell terminals (apposed by CALB+ terminals) and inhibitory terminals that come from other neurons (not apposed by CALB+ terminals). In the revised manuscript we now report the % of gephyrin puncta that **are apposed** to CALB+ terminals as a measure of input from Purkinje cells. We conclude that, since the percentage of Gephyrin puncta apposed to CALB+ terminals remains unchanged, this could be due to a combination of terminals from Purkinje cells and inhibitory interneurons. We hope this increases the clarity and readability.*

- It is not clear how appositions were quantified and what threshold was used to determine if gephyrin puncta faced or did not face a calbindin-positive terminal. Please explain and justify this analysis.

Gephyrin staining leads to clear, discrete puncta that can easily be distinguished from background fluorescence - there was no threshold necessary in order to identify them. Counting was done manually using ImageJ. It is generally very clear when a gephyrin puncta directly neighbours a CALB+ terminal, visible as fluorescent pixels side by side. We have now edited the Methods section to clarify this point.

- In Figure 4H-I, the authors show that spontaneous Purkinje cell firing rates are higher in conditioned animals than in naïve ones. They interpret this in the context of the structural

changes they observe in ponto-nuclear projections (Paragraph starting on line 292). This seems like a strong overinterpretation of their results, as no direct measurements of structural changes in the cerebellar cortex have been made. Moreover, it is not obvious (to me) that an increase in mossy fiber input would necessarily lead to higher rates of Purkinje cell spiking. The cerebellar cortex goes to great lengths to balance excitation and inhibition, so it is equally plausible that this 'extra' mossy fiber input is cancelled by cortical interneurons before it reaches Purkinje cells. I would suggest tempering this claim or demonstrating directly that mossy fibers drive stronger Purkinje cell spiking after learning.

We agree with the Reviewer. We indeed did not have direct evidence that the cause for this lies in changes in mossy fiber inputs to the cerebellar cortex. We have further investigated this by quantifying the mossy fiber terminals in the granule cell layer of the cerebellar cortical eyeblink region, but we did not find any relevant change in these inputs (see new Figure 2j-k). We have therefore decided to remove the results regarding the changes in spontaneous Purkinje cell activity, let alone to speculate on the data.

- In Figure 5, the authors demonstrate nicely that pons stimulation can evoke eyelid closures after training but has essentially no effect pre-training. This interesting result begs several questions: Related to point 1, do the authors observe whole-body defensive movements when they stimulate the pons after training (as in Figure 5)?

This is an interesting point. In these particular experiments we only recorded the eyelids, so unfortunately we cannot answer this specific question. However, given the whole-body movements over the course of the new eyeblink conditioning vs. pseudo-conditioning experiments we did (see the new Supplementary Fig. 7), it is certainly reasonable to speculate on this. We have now adjusted the Discussion in line with our new findings on this topic.

Would the results of this experiment look different if the photostimulation was done directly in the DLH or IntA regions of the cerebellar nuclei?

This is definitely a valid question. We have started a collaboration with Megan Carey and Catarina Albergaria to address this question, as they already had extensive expertise and data on this topic. It is clear from these new experiments that optogenetic mossy fiber stimulation at the level of the anterior interposed nucleus leads to predictive eyelid movements over the course of learning. Importantly, the timing of the responses is adequate and adjustable, whereas the potential for modulation by locomotion is absent. These data are now added.

Does pons stimulation in trained mice result in stronger spiking responses in CN neurons?

There is strong, albeit partially indirect, evidence that this is indeed the case. In our revised manuscript we now show that larger conditioned eyeblinks can result from optogenetic stimulation of mossy fibers in the CN (see the new Figure 1c). We also show that learning leads to increased pontocerebellar mossy fiber inputs to the CN (see new Figure 2). Moreover, optogenetic stimulation of pontine nuclei neurons leads to short-latency spike facilitation in the majority of the responding neurons (see new Supplementary Figure 3c), which most likely results from direct excitatory mossy fiber input to the CN, since activation of granule cells in GABRA6-

ChR2 mutant mice leads predominantly to inhibition in the CN (see new Supplementary Figure 4). Furthermore, previous juxtosomal recordings from CN neurons across many days of training (Ten Brinke et al., 2017 eLife - Figure 1M) also indicate that the majority of CS-responding neurons in the CN show increasing spike facilitation coinciding with the occurrence of CRs. In addition to showing all the new datasets, we now also address this question and the related findings in detail in the Discussion.

- Abstract – This sentence is not grammatically sounds: ‘Here we studied these changes in inputs using whole-cell and juxtosomal recordings of cerebellar nucleus neurons in awake mice during eyeblink conditioning and doing post-learning structural analysis.’ I think the easiest fix is to add the word ‘by’ between ‘inputs’ and ‘using’.

This has been corrected in the revised manuscript.

- ‘EPSP/IPSP neuron’ nomenclature – it seems presumptuous to use these names to refer to the neurons activated and suppressed by the CS. As the authors themselves point out, there could be different mechanisms that lead to the membrane dynamics they observe (e.g. disinhibition – line 94). I would suggest the authors switch their naming terms to something more descriptive like CS-activated and CS-suppressed neurons.

We thank the Reviewer for pointing this out. In line with your suggestion, we now call neurons showing CS-evoked Vm depolarizations ‘CS-activated’ and those showing Vm hyperpolarizations ‘CS-suppressed’ throughout the manuscript.

- Line 285 – technically, ‘VGLUT1 expression’ (i.e. mRNA) was not measured here. Please change to a more accurate term like ‘VGLUT1 puncta density’

This has been corrected.

- Figure 4 and other places – it would be more intuitive if the authors referred to the left and right hemispheres as ipsilateral and contralateral to the side of the conditioning. This is the more relevant comparison that left and right, and exhibiting the data in this way does not require a reader to remember the side on which the animals were conditioned.

We now refer to the left and right hemisphere as ipsilateral and contralateral relative to the US, respectively.

- Figure 5F – It looks like this is an average across animals (according to the legend) but there are no error bars on these traces. Please add these.

This has been corrected.

- Line 735 – coordinates should be in millimeters, not micrometers.

This has been corrected.

Reviewer #2

In this manuscript Broersen et al utilize the technique of whole cell and cell attached single cell recording from the cerebellar nuclei of mice while they perform a conditioned eyeblink behavior. Additionally, the authors use antibodies to examine for changes in excitatory and inhibitory connections in the CN after conditioning, and they use optogenetic activation of pontine nuclei to determine changes in mossy fiber input after conditioning.

The most significant insight we do get, albeit with very small N's and on somewhat shaky statistical grounds, is that depolarization in DHP and/or IntA neurons during the CS in conditioned animals precedes spike rate changes and is enhanced after conditioning. In addition, a linear decoder shows that this early activity has modest predictive information on the ensuing behavior.

We thank the Reviewer for the positive and constructive feedback.

- In vivo whole cell recording from deep structures in behaving mice is an incredibly difficult technique, which explains the really low sample number of neurons obtained (6 from naïve mice, and 8 from conditioned mice). The authors do themselves a bit of a disservice by then being very lax about rigor in their statistical analyses of these numbers. It is never mentioned how many mice these neurons are from, which is a critical piece of information. Nor do they show which ones are from the DLH or IntA nucleus, which is important in light of differential antibody staining findings between the 2 nuclei. A number of statements about changes seen are not supported by statistical significance, which may be unsurprising given the small N's. Scientific rigor would require that for each results statement, the n of neurons, the n of mice, and the nuclei involved would be stated.

We thank the Reviewer for bringing these points up. To address these issues we have taken the following steps. 1) We now consistently report the number of neurons and the number of mice. 2) We have added more neurons to the whole-cell recordings dataset, now totalling n = 15 neurons for both groups (naive N = 11 mice; cond N = 12 mice). We have redone the statistical analysis, which now has more power. With the new (larger) dataset, we still find the same overall findings.

Regarding the position of neurons in the subnuclei: unfortunately, filling the recorded neurons with neurobiotin was often not feasible given the short duration of the recordings, despite our continuous attempts to identify where the recorded neurons were located. It is therefore not possible to indicate the specific subnucleus, other than that the successful track was always leading to the AIP, including the DLH (although one can argue about this, we do consider the DLH as part of the AIP; see De Zeeuw and Ten Brinke, 2015 CSHP). Moreover, it should be noted that the recording location was always established as the eyeblink region with fluorescent tracer labelling, microstimulation inducing eyelid movements and recording spike activity (see e.g., Figure 3b). We now describe the location consistently as AIP, and we note early on in the manuscript that we consider the DLH as part of the AIP, while referring to De Zeeuw and Ten Brinke, 2015. Moreover, we discuss the limitations of our locus determination in

the Discussion.

- One of the major advantages of whole cell recordings is that one could discern excitatory from inhibitory postsynaptic responses by injecting a bias current for some of the traces, or perform a voltage clamp analysis. Such methods have for example been utilized by Destexhe et. al. The authors do not perform this test and instead call any depolarizing Vm deflection an EPSP, although this could either reflect disinhibition by reduced PC input or increased excitation by MF input. While this is acknowledged by the authors in some remarks, it is not following the usual definition of an EPSP, which should be due to excitatory input, not disinhibition. As a consequence, the answer that would be most interesting to get, namely if MF excitation to CN during the CS in response to conditioning is responsible for the added depolarization seen, is not given.

While we agree with the reviewer that insight in the excitatory and inhibitory synaptic inputs during the task would be a valuable addition to the manuscript, unfortunately the technical limitations did not allow us to perform these voltage-clamp recordings. Recordings were generally short, with a median duration of 149 seconds, and recordings were lost or lost quality as soon as the mouse started walking on the treadmill. During our current-clamp recordings we aimed to record trials with and without CR or EO, preferably with varying levels of CR/EO amplitude. We also administered US trials, all while using an inter-trial interval of at least 8 seconds to not overload the mouse and match the training regime, and allowing the membrane potential of CN neurons to recover before giving the next stimulus. To make any meaningful comparison we required multiple trials per trial condition and this could usually be done in the limited amount of time per recording. However, it turned out to be technically too challenging to record the same neuron both in current- and voltage-clamp for a sufficient number of trials. Indeed, we did try to change the holding potential to depolarized potentials in order to isolate inhibition, but this often resulted in unhealthy neurons.

We chose to perform current-clamp recordings, because they reflect precisely how the neuron would behave, without our interference, during the task. This includes integrating the total sum of inputs and observing how this leads to spiking activity. This by itself provided a wealth of information about the specific characteristics of the neuronal responses. If we had changed the contents of our intracellular solution (e.g., high chloride, cesium-chloride), it would have prevented us to acquire current-clamp recordings. Thus, we chose for this study to get a robust dataset on the current-clamp recordings with a sufficient n and N, and we envision a new study with voltage clamp recordings for the future. The pro's and cons of our approach are now discussed more upfront (see Discussion). Finally, along the same vein, please note that in line with the suggestion by Reviewer 1, we now call neurons showing CS-evoked Vm depolarizations 'CS-activated' and those showing Vm hyperpolarizations 'CS-suppressed'.

- The optogenetic stimulation of the pontine nuclei also affects both input to the cerebellar cortex and the cerebellar nuclei, not allowing any disambiguation of the relative pathway significance in this behavior. Optogenetic inhibition of mossy fiber terminals from the PN in the CN would have been a much stronger experiment to support the functional role of excitatory input to these neurons in conditioned depolarizations.

We fully agree. We have now added new experiments where MFs are stimulated directly in the CN in the context of learning, thereby addressing this concern; this stimulation is sufficient to induce timing-dependent EBC (see new Figure 1). In addition, we performed juxtosomal recordings from the CN during PN stimulation and demonstrate that most CN neurons show short-latency (< 10 ms) spike facilitation (Supplementary Figure 3c). Since optogenetically activating granule cells in the GABRA6-ChR2 transgenic mouse predominantly led to inhibition of CN cells as a result of increased Purkinje cell simple spike activity (Supplementary Figure 4), this short-latency spike facilitation in the CN following PN stimulation likely results from direct excitation of MF collaterals in the CN. Moreover, we co-injected Chronos with the inhibitory eArch3.0 in the bilateral pontine nuclei and we show that increased eyelid closures over the course of conditioning are only seen during blue light activation of MFs, but not during inhibition of MFs (Supplementary Figure 3e-f). Thus, the effect is specific to optogenetic MF activation and the eyelid movements are not general reflexive movements to light onset. As noted by the Reviewer, the time course of the eyelid response, which starts right after light onset, indeed further argues in favour of MF activation innervating the CN causing the eyelid movements. Thus, there are multiple lines of evidence indicating that optogenetic activation of the direct excitatory input to the CN is likely responsible for the eyelid closures.

- Table 1 should include columns, for animal ID, and nucleus recorded.

Table 1 now includes the mouse ID's. As discussed above, all data are from AIP.

- Line 77/78. Example of results statement without significance due to low n.

This has been corrected.

- Line 96/97. I don't understand how CS responses can be later in CS only trials than in CS/US trials, given that the mice have no information on whether a US will follow, or do they? The Methods are not clear on the sequencing of trial types during recording sessions (lines 581-594).

We agree. With regard to the CS responses we have removed the analysis of CS-only trials, because of the low number of trials for this condition. Conditioned mice received more paired trials than unpaired CS or US trials to avoid extinction during the experiment. As a result, the number of CS-only trials is small compared to CS-US paired trials. The Methods section now includes a description of the trial types during recordings, and we highlight that the CS-only trials yielded to few responses for statistics in this respect.

- Line 102. Example of missing statement of N's. "For all neurons for which we recorded both CR+ and Cr- trials ..."

This has been corrected.

- Line 104/106. Reporting results from a single neuron becomes really underpowered.

The Reviewer is correct. The dataset contains a low number of neurons that showed CS-evoked Vm hyperpolarizations. In the revised manuscript we particularly focus on neurons showing CS-evoked Vm depolarizations, as these are the responses most often observed in our dataset and these are thought to underlie increased spiking activity that leads to activity in the downstream motor nuclei controlling the eyelids.

- Line 119/120. Another effect statement without N's. (numbers of mice, numbers of neurons, which nucleus?)

This has been corrected.

- Line 191/192. With a p-value of 0.031 we learn that the contralateral hemisphere has increased VGLUT1+ puncta compared to the ipsilateral (which carries the conditioned eyeblink response). This counterintuitive result may be spurious given the small N's (in this case of mice processed with antibodies).

In the revised version new mice have been trained (conditioning as well as pseudo-conditioning) and synaptic puncta have been quantified in these mice. This has led to more robust quantifications - we now show that VGLUT1+ excitatory puncta are bilaterally increased in the CN and that no significant difference between both sides exists. In addition, to isolate VGLUT1+ MFs, we have made fluorescent tracer injections directly in the pons bilaterally. Here we also find that YFP+ varicosities are increased bilaterally, further supporting our VGLUT1+ findings.

- How many statistical tests were made in total, and may this outcome be a result of p-fishing? A Bonferroni or similar adjustment should be made for the number of total tests attempted.

Please note that we only made the statistical comparisons to test the hypotheses that we made. Adjustments for multiple comparisons are done adequately. For the comparison in question, this was a one-way ANOVA with Tukey's multiple comparison post-hoc test.

Reviewer #3

This manuscript by Broersen and colleagues aims to examine structural and synaptic changes in the connection from mossy fibers to cerebellar nucleus (CN) neurons following eyeblink conditioning, a form of associative learning that is dependent on the cerebellum. The authors use a variety of techniques including in vivo whole cell patch clamp and juxtасomal recordings during awake behavior, immunohistochemistry, and optogenetics to look for changes in this pathway following learning. They find some evidence that is suggestive of learning-induced remodeling of mossy fiber--but not climbing fiber--inputs to CN, including increased EPSPs in response to the conditioned--but not unconditioned--stimulus, increased expression of a marker of mossy fiber--but not climbing fiber--terminals, and a progressive increase in eyeblink size driven by optogenetic stimulation of pontine mossy fibers during conditioning. The mossy fiber to CN pathway is an understudied and potentially important site of plasticity underlying cerebellar-dependent learning so the topic addressed by the paper should be of high interest to the cerebellar community. In addition, the attempt to link structural changes with synaptic properties examined electrophysiologically increases the potential impact of the paper. However, the study suffers from several conceptual and methodological issues that prevent the experiments from supporting the conclusions and detract from its overall significance to the field.

We thank the Reviewer for the constructive feedback.

- The number of neurons recorded by whole cell patch clamp does not seem sufficient to support the conclusions. I understand that these are very difficult experiments to perform, but the decision to record from both the ipsilateral and contralateral CN in conditioned mice would seem to have unnecessarily reduced statistical power given what we know about the *ipsilateral* control of movement by the cerebellum. If the goal was to look for structural and synaptic plasticity in the neurons that are responsible for driving the learned behavior it would have been better to focus on the ipsilateral (to the air puff) IntA and DLH, which are more established as being essential for generating learned eyeblinks (however, see related comment below regarding the histology results).

We agree with the Reviewer that recording from both the ipsi- and contralateral hemisphere introduces an extra variable that may affect statistical power. However, there were multiple reasons why we recorded from both hemispheres; 1) several papers show that processing of such movements occurs bilaterally in the cerebellar nuclei (e.g., Soteropoulos and Baker, 2008 J Physiol; Cui et al., 2000, Neuroreport) and that cerebellar lesions also affect the contralateral side (Immisch et al., 2003, Neuroreport). More specifically to EBC in mice, changes in MF inputs to the CN after learning also occur bilaterally (Boele et al., 2013 J Neurosci.). Thus, it is likely that both hemispheres are involved in EBC. 2) from an ethical perspective, we wanted to record as much data as we could from the same conditioned mouse. Inserting a glass electrode into deeper structures (i.e., 1800 μm from the pia) to reach the nuclei damages the brain tissue, which in practice means that after 5-6 electrode penetrations the chance of obtaining a successful whole-cell recording drastically decreases. These mice were handled/trained for

multiple weeks and had undergone two surgeries - we cannot justify it to not use the other hemisphere. Furthermore, when we tested the relation between laterality and CS-evoked Vm depolarizations, we did not identify a significant difference (cond: $P = 0.77$, naive: $P = 0.571$, unpaired t-test).

- Based on Fig. 1f there only appear to be about 2-3 neurons that have consistent EPSPs during the ISI in conditioned mice. These numbers are unlikely to allow any kind of meaningful statistical tests and render statements about the relative changes in EPSPs for conditioned and unconditioned stimuli inconclusive. The pooling of trials does not alleviate this issue because it's not clear what proportion of trials is coming from each neuron and how this is accounted for in the statistics. The low "n" is especially problematic because the whole cell patch clamp experiments are a major part the paper and arguably the biggest novelty.

We agree. In the revised manuscript we have added additional neurons to our original dataset, which now comprises of $n = 15$ neurons per group. ROC analysis indicated that 12/15 (conditioned) and 11/15 (naive) neurons responded to the CS. We are confident that with this larger dataset this issue is resolved.

- The structural changes revealed by VGLUT1/2 expression are hard to interpret based on what we know about the eyeblink-controlling regions of the CN. Given the importance of the ipsilateral IntA/DLH in driving the learned eyeblinks, it's hard to make the case that the changes in mossy fiber input to the contralateral CN are driving the learned responses. The authors cite Heiney et al, 2021 to say that movements of multiple body parts are learned during eyeblink conditioning but this paper actually shows that all of the movements are still driven by the ipsilateral IntA.

The results that emerge from the new more extensive data-set on the VGLUT1 and MF quantifications indicate that changes in afferents (excitatory as well as inhibitory) occur bilaterally. This resolves the issue of laterality in indicators for plasticity.

- It needn't be the case that the only plasticity relevant to the learned behavior is happening in the "eyeblink controlling" part of the CN but it does seem problematic if there is no evidence of plasticity there. In my mind, this substantially decreases the significance.

While we agree with the Reviewer on this point, we would like to note that plasticity changes during EBC may come in different forms. For example, structural increases in numbers of inputs (this study, as well as Boele et al., 2013 J. Neurosci.) may be complemented by changes in intrinsic plasticity of CN cells (Wang et al., 2018, PNAS), changes in synaptic transmission without structural changes, or changes in activity from inputs (e.g., Ten Brinke et al., 2015 Cell Reports). Despite the limitations of the current approaches, we have still been able to detect changes in excitatory and inhibitory inputs. Moreover, these findings are now further supported by the increase in YFP MF varicosities in the CN. We hope that this addresses the Reviewer's concerns.

- There is also little attempt to relate the structural changes in mossy fiber innervation to the

electrophysiology. Were the neurons with large EPSPs during the ISI recorded in the same region/side as the increased VGLUT1 was found?

The whole-cell recordings during eyeblink conditioning enable us to directly measure the effect of changes in inputs (excitatory as well as inhibitory) on the membrane potential of CN neurons. This, in combination with our data showing structural increases in mossy fiber inputs as well as Purkinje cell inputs, is in our view a suitable approach to test a direct relation between physiology and anatomy. Whole-cell recordings are better suited than juxtosomal recordings in this respect, because increases in inputs may lead to changes in subthreshold activity, but not necessarily lead to changes in spike activity. With respect to the laterality of the VGLUT1 densities in relation to the neuronal responses, we found neurons showing Vm depolarizations in both hemispheres. Our newly added data further suggest that changes occur bilaterally. We have now highlighted this better in the revised manuscript.

What about the juxtacellularly recorded neurons with increased spiking during the ISI? This is glossed over in the presentation of the results. The data would be easier to interpret as a whole if more effort was made to tie together the results from the various experiments in the paper.

We found CS-responding neurons in both hemispheres during the juxtosomal recordings. In the ipsilateral and contralateral hemisphere we found that 52.7% (n = 55 neurons) and 33.3% of the neurons (n = 33 neurons) responded positively to the CS, respectively. This information has now been added to the revised manuscript.

- Related to these points, the learning in many of the conditioned mice appears to be poor (the insets in many figures show average CR amplitudes of 10% full closure, which is the minimum threshold the authors use to define a CR). If the mice didn't learn well how can we interpret any of the findings from the electrophysiology and histology experiments, especially given the small sample sizes and relatively large responses of the neurons in naive mice? The study seems generally underpowered for the conclusions.

This comment appears to be based in part on a misunderstanding. The learning of conditioned mice during the training phase was generally strong (see Supplementary Fig. 2) and we made sure that mice were only subjected to whole-cell recordings or histological quantification when they showed a high performance (on average ~75% CR). The amplitude of the resulting CRs was variable, but on average higher than 30% of the UR. The insets featured in some of the figures (e.g., Figure 3f and 4a) represent the eyelid traces only of the trials during the whole-cell recordings. These recordings could only be made after a bilateral craniotomy surgery under isoflurane anaesthesia (Methods), where mice had ~2 hours to recover. The reason for this is that once the craniotomy is made, the quality of the brain quickly decreases to a level that makes whole-cell recordings impossible. Thus, limiting the time between the craniotomy and recordings is essential for success. The consequence is that mice did not perform as well during the recordings as they initially did, which is reflected in the lower CR amplitude value. Nevertheless, a smaller CR amplitude value does not mean that the findings cannot be interpreted, although we do acknowledge that larger CRs may be associated with stronger synaptic responses in the CN.

- The juxtosomal recordings do not appear to add much to what is already known in the literature. The main value of including these recordings would seem to be to relate them to the whole cell recordings but the connection between these two is underdeveloped.

The juxtosomal recordings add in that they confirm that we are targeting the eyeblink-processing region of the CN in the electrophysiological experiments. Second, they show that the largest difference in spike activity contrasting CR vs. NR trials is observed in the last period of the CS-US interval, closely reflecting the time course of Vm depolarizations (Figure 3b). Furthermore, the juxtosomal recordings are also used for the decoding analysis, showing that precisely those time periods are most informative to distinguish CR vs NR trials.

- Were cell attached recordings ever made before going whole cell, which might allow a more direct cell-by-cell comparison of input-output? As is, there is a reliance on a population analysis of cells recorded using each method but since the n for the whole cell recordings is so small it's hard to appreciate the connection to the juxtosomal recordings.

For obtaining a successful whole-cell recording in an awake mouse it is crucial to limit the time between cell-attached and break-in, as movements of the mouse may decrease the giga-seal quality. Since whole-cell recordings were our priority, we did not acquire cell-attached recordings before going whole-cell. However, we expect that these findings would not be different from the juxtosomal recordings as presented in this manuscript, since the distribution of responses is similar to the whole-cell data (note that the higher percentage of no response neurons in the juxtasomally-recorded dataset could represent neurons that show synaptic processing, but no spike changes). With the increased number of whole-cell recorded neurons in the revised manuscript, we hope that the remainder of the Reviewer's concerns on this point is resolved.

- The decoder results aren't particularly enlightening given the small sample sizes because presumably a decoder would perform better if it were "listening" to more neurons than were included in the sample (especially for the whole cell recordings).

We have now added additional cells to the decoder, increasing the number of trials. Furthermore, instead of comparing parts of the dataset based on the CS responses, we now compare the conditioned versus the naive neurons at a group level, which appears to be informative and relevant.

- The US coding results suffer from the same issue. It's hard to prove the negative when there is a hint of a difference between groups that could become significant if more cells were included. There is no power analysis provided to suggest that the sample size is appropriate to answer the question.

We agree with the Reviewer. As mentioned, we have added additional neurons to the dataset. However, given the variability in UR profiles and US-evoked Vm responses, we cannot discount the possibility that we are underpowered for this comparison. We have now added a power analysis in the text to reflect this.

- There is no discussion of alternative plasticity mechanisms. The authors focus solely on (presynaptic) mossy fiber structural plasticity and attempt to link it to the observed CS-driven EPSPs in CN neurons but other potential plasticity mechanisms, including post-synaptic changes in CN neurons, are not considered or discussed.

It is hard to determine what types of post-synaptic changes in CN neurons the Reviewer is referring to. We have investigated structural changes in VGLUT1 mossy fiber input, Purkinje cell inhibitory input, as well as baseline activity of Purkinje cells (although the PC analysis has been removed; see comments made by Reviewer 1). Changes in intrinsic excitability of CN have been considered, but we could not find significant differences, which might be partly due to the diversity of cell types in the CN as well as our limited sample size. The synaptic transients we measured are reported as they are, and they may reflect the total sum of plasticity affecting the recorded neuron. Yet, as suggested by the Reviewer, we now added a paragraph in the Discussion about what other plasticity mechanisms may take place.

- The optogenetics experiment as performed cannot answer the question it sets out to answer because stimulating in the PN will also activate mossy fibers projecting to the cerebellar cortex, which could lead to larger eyeblinks due to plasticity within the cortex regardless of changes in the CN. In addition, the experiment is missing a couple of critical controls, which makes it impossible to interpret. There needs to be a control for increased Chronos expression over time, which could also explain the results. A pseudoconditioned group would be one possibility but perhaps there are others as well. It also needs a GFP or similar control to make sure the mice aren't just blinking to the laser light as they learn (since the CS is also a light), although depending on the time window analyzed this may not be necessary.

This Reviewer is correct to point out several caveats of the optogenetics experiment. As also highlighted above, we made several additions to the article to address these concerns. First, we performed juxtosomal recordings from the CN during PN stimulation, demonstrating that most neurons show short-latency (< 10 ms) spike facilitation. Second, since optogenetic activation of granule cells in the GABRA6-ChR2 transgenic mouse predominantly led to inhibition of CN cells as a result of increased Purkinje cell simple spike activity (Supplementary Figure 4), this short-latency spike facilitation in the CN following PN stimulation likely resulted from direct excitation of MF collaterals in the CN. Moreover, we co-injected Chronos with the inhibitory eArch3.0 in the PN and we show that increased eyelid closures over the course of conditioning are only seen during light-activation of MFs, but not during light-inhibition of MFs (Supplementary Figure 3e-f). Thus, the effect is specific to optogenetic MF activation and the eyelid movements are not general reflexive movements to light onset, but argue that the MF activation causes the CRs. Last but not least, we added optogenetic activation of MFs at the level of the CN, which lead to larger eyelid closures over the course of conditioning, while maintaining temporal flexibility (Fig. 1c).

- Given the small n for many of the experiments wouldn't it be more appropriate to report medians and ranges rather than means and SEMs for the summary statistics?

With the added datasets and n's for almost all statistical comparisons, and the use of predominantly parametric tests, we choose to report means and SEMs as done in the revised manuscript.

- When using a paired t-test with missing data from one group, wouldn't the "n" be the lesser n of the two groups, i.e. only those cells without missing data?

It is hard to determine which statistical comparison the Reviewer is referring to here. However, we have updated all paired t-test results to reflect the correct 'n'.

- Figure 1e: It's hard to see the eyelid traces when the voltage traces are displayed on top of them.

In the revised version we have restructured the figures.

- i/j: It would help to show different neurons with different markers to appreciate the relative proportions.

In the revised version we have restructured the figures. Some of these panels are now featured in Supplementary Figure 6. Given the amount of data points we choose to not color-code the neurons, since the clarity of these figures will decrease. However, we have now added individual slope lines for each neuron.

- k: On the printed page, it's hard to tell the difference between the markers for the "cond IPSP" and "naive" groups.

In the revised version we have restructured this figure panel, by increasing its size and adding additional neurons (see new Figure 4i). These adjustments made the markers easier to distinguish.

- Figure 4e/f: Wouldn't it be more direct to quantify Gephyrin+ puncta that ARE apposed to CALB+ boutons as "PC" terminals rather than the way it was done? Is there a reason why this can't be done? The reasoning seemed convoluted.

We have rewritten the text regarding the Gephyrin+/CALB+ puncta. We thank the Reviewer for this suggestion.

- There are several references that appear inappropriate/incorrect. For example, ref 23 on page 2 line 44, ref 33 on page 3 line 87, and ref 50 on page 7 line 220 are cited for things that they don't appear to have anything to do with, but there may be others as well.

Several major adjustments to the manuscript text have been made in the revised manuscript, including the references the Reviewer is referring to; e.g., ref 23 on page 2 and ref 33 on page 3 have been reallocated, and ref 50 on page 7 has been deleted.

- The logic of the following statement is not clear to me (page 7 line 218): "The increased

density of VGLUT1+ MFs in the DLH after conditioning may suggest that conditioning is also facilitated by an overall increased MF drive to the cerebellar cortex." Can this be elaborated?

This section of the Results has been updated to include new results. We now show that VGLUT1+ MFs are increased at the level of the CN, but not in the granule cell layer (quantification of MF rosettes). The hypothesis that we referred to in this statement, i.e., that increased MFs in the CN could coincide with increased mossy fibers in the cerebellar cortex, has therefore been falsified. We have adjusted the manuscript text accordingly.

REVIEWER COMMENTS

Reviewer #1 (Remarks to the Author):

Summary:

In this revised manuscript, Broersen, Albergaria, Carulli and colleagues have made substantial additions and alterations that have improved the study. These changes have been positive both in terms of the scientific findings as well as the clarity of the study. However, given these substantial alterations, I have some new suggestions related to the new experiments. I have tried to limit my suggestions to ones that don't require new experiments. Once these are addressed, I would support the manuscript's publication.

Points that still need to be addressed:

1. The reference list is a mess – the numbering in the main text has not been updated to reflect the added references in the revision.
2. I would encourage the authors to minimize the editorial language in their manuscript. As an example, the authors discuss 'surprisingly well-timed movements' on lines 62-63 in the introduction. The word surprising is redundant with the notion of doing science and discovering new things, so it's probably better to just leave it out. This is meant to be an illustrative example of a larger point, so if there are other such phrases in the manuscript, I would encourage the authors to edit accordingly.
3. At the end of the introduction, the authors discuss 'both structural and physiological changes' as if they are independent things that would not impinge on each other. These are likely to be related, so the phrasing of this statement should reflect this possibility.
4. For clarity, I would suggest that the authors make it explicit that the DREADDs experiments are done in mice expressing the receptor constructively under the control of the GABRA6-Cre line (rather than for example by viral expression specifically in the eye-blink region of the cerebellar cortex). This means that all GrCs are likely affected (as well as any other cells that have the Cre). I also think it would be appropriate to discuss the specificity/off-target effects of their expression strategy.

5. New figure 2:

- The normalization in panels C-D is somewhat confusing. The methods (lines 1301-2) describe normalizing within batches of pseudoconditioned mice, but it was unclear (to me) how these batches were defined. It also seems appropriate to have the 100% tick labeled in such a normalized bar graph, since this is what we are comparing to.
- It is not clear from panels E-F that the x-axis is normalized. This should be clear.
- I personally find the y-labels in panels I and O (i.e. n/mm) to be confusing. Please make these labels more informative.

6. Discussion of GrC stimulation experiments (lines 211-218): The authors use excitation and inhibition to mean increases and decreases in spiking. Please use the more accurate terms.

7. The authors describe (lines 302-312) that Vm in some neurons correlates with eyelid movements, and it does not in others. Then they say that in the (pre-selected) correlated group, the population exhibits a particular set of encoding properties and other group does not. But of course, this was pre-destined by the selection criteria. Please rephrase.

8. The authors perform some truly heroic experiments in recording membrane dynamics during eye-blink conditioning (detailed in Figure 3-4). Only a subset of these neurons exhibit membrane dynamics that one would expect from the spike responses shown in previous studies, but they are actually in agreement with the prevalence of response types shown in this study (i.e. Figure S9). I think this is a point that is worth emphasizing a bit more.

9. It seems to me that the analysis in Figure 5 is doing a disservice to the data that has been collected. I suspect that if the authors applied some sort of weighting function, rather than simply pooling, before applying LDA, they would get better performance of their decoder. Surely, this information exists within the population (e.g. Figures 4B, S6B, S9C, among others).

10. In the discussion, the authors imply that their chemogenetic experiments result in a complete removal of the cerebellar cortex's influence. This is not strictly true – e.g. Purkinje cells probably still get climbing fiber input and fire complex spikes in response to the US. The

authors should qualify accordingly.

Reviewer #2 (Remarks to the Author):

The thoroughly revised manuscript includes many new experiments and data points that significantly strengthen the conclusions drawn. Specifically the new direct optogenetic stimulation of mossy fiber terminals in the AIP closes an important gap that was apparent in the first version of the manuscript, and the added anatomical data result in a more solidified and bilateral finding of mossy fiber terminal changes in the AIP due to conditioning. The data on which the conclusions rest now look sound. Further, the removal of using EPSP/IPSP verbiage makes the interpretation of whole cell recordings more clear and commensurate with the measurements taken. The conclusions now drawn clearly add some important insights to mouse cerebellar eyeblink conditioning with respect to the involvement of mossy fiber input to the AIP. They also illustrate the complexity of evoked movements both in naïve and in trained mice in a way that is helpful to further development of this field. I only have a few remaining minor questions / comments.

Minor Comments.

The optogenetic CS for 200 or 500 ms eyeblink conditioning lasts throughout the entire delay period unlike a classic short sensory CS (Fig. S1). The authors should make this difference explicit in their interpretation of results with respect to what it means for the control of learning the timing of a CS-US interval. Would the mice have learned a 500 ms CS-US pairing if the CS similarly only lasted 200 ms as in the 200 ms pairing?

Lines 164/165. 1292-1310. Please state if the investigators were blinded as to which type of animal was analyzed. Without blinding it is not clear if this method of scoring varicosities does not allow for a subjective factor, like unconscious bias.

Lines 474-477; 488/489. It seems tenuous to attribute the US responses in the AIP to CF input. There would be also mossy fiber input, including sensory feedback elicited by the movement. For a close association with CF input, the relative delay of CF input and

depolarization should be shown to match. Does CF input also show a preceding CR influence like the Vm encoding of the US? Otherwise the Vm encoding would not seem to be a direct outcome of CF activation.

Reviewer #3 (Remarks to the Author):

In this revised manuscript, the authors have addressed most of the initial critiques by all reviewers and added an impressive amount of new data and experiments. As a result, it is a much stronger paper. In particular, the inclusion of the direct mossy fiber stimulation in the AIP and the corresponding finding regarding timing of MF-AIP CRs is interesting and challenges a long-standing model in the field, which greatly increases the significance. The restructuring of the paper as a result of the newly added experiments as well as the inclusion of a larger number of neurons for the whole cell patch clamp experiments alleviate my concerns about the study being under-powered.

I have no further major concerns. However, if the manuscript is accepted, I urge the authors to double-check all references before submitting the final version as I again noticed many apparently incorrect references. I cannot provide an exhaustive list but as an example most of the references in the third paragraph of the Introduction do not appear to be correct.

REVIEWER COMMENTS

Reviewer #1 (Remarks to the Author):

Summary:

In this revised manuscript, Broersen, Albergaria, Carulli and colleagues have made substantial additions and alterations that have improved the study. These changes have been positive both in terms of the scientific findings as well as the clarity of the study.

We thank the Reviewer for their positive feedback.

However, given these substantial alterations, I have some new suggestions related to the new experiments. I have tried to limit my suggestions to ones that don't require new experiments. Once these are addressed, I would support the manuscript's publication.

Points that still need to be addressed:

1. The reference list is a mess – the numbering in the main text has not been updated to reflect the added references in the revision.

Due to an issue with the reference manager, the references were indeed not correctly listed. We have now updated and corrected all references accordingly. We apologize for this mistake.

2. I would encourage the authors to minimize the editorial language in their manuscript. As an example, the authors discuss 'surprisingly well-timed movements' on lines 62-63 in the introduction. The word surprising is redundant with the notion of doing science and discovering new things, so it's probably better to just leave it out. This is meant to be an illustrative example of a larger point, so if there are other such phrases in the manuscript, I would encourage the authors to edit accordingly.

We agree with the Reviewer. We have removed the word 'surprisingly' from this specific sentence. We have also made several other adjustments throughout the manuscript to minimize the editorial language.

3. At the end of the introduction, the authors discuss 'both structural and physiological changes' as if they are independent things that would not impinge on each other. These are likely to be related, so the phrasing of this statement should reflect this possibility.

We agree with the Reviewer that structure and physiology are tightly linked. To better reflect their relation, we have adjusted the phrasing as follows (line 82-84): "Together, our findings raise the intriguing possibility that there is the capacity for sensorimotor learning and temporal coding within the CN themselves, carried by a synergy of structural and physiological changes at the level of individual nuclei neurons."

4. For clarity, I would suggest that the authors make it explicit that the DREADDs experiments are done in mice expressing the receptor constructively under the control of the GABRA6-Cre line (rather than for example by viral expression specifically in the eye-blink

region of the cerebellar cortex). This means that all GrCs are likely affected (as well as any other cells that have the Cre).

We agree with the Reviewer that this point could be made more explicitly. We have edited the text accordingly, which now reads (line 112-118): “ We further examined the specificity of optogenetic stimulation of MF terminals in the AIP vs. CTX by combining optogenetic MF CS stimulation in AIP with the use of inhibitory Designer Receptors Exclusively Activated by Designer Drugs (DREADDs)^{34,35} expressed in cerebellar granule cells, the targets of MFs in the cerebellar cortex (Fig. 1f), under the control of the Gabra6 promoter³⁶. These mice expressed ChR2 in MFs as well as the inhibitory DREADD hM4Di in granule cells throughout the cerebellar cortex, as confirmed by immunohistochemistry (Fig. 1g; mice from here on termed MF-ChR2-gc-DREADD).”

I also think it would be appropriate to discuss the specificity/off-target effects of their expression strategy.

In addition to the section of the Discussion that mentions the limitations of our chemogenetic approach, we added the following text to the Methods, in the section describing these experiments (lines 1233-1236): "Specific DREADD expression in cerebellar granule cells was confirmed histologically. Since it is difficult to ensure complete cell-type specificity in BAC transgenic Cre lines, key experiments were replicated in another granule cell-specific Cre line with nonoverlapping patterns of expression outside of granule cells."

We also note that while the Funfschilling and Reichardt Cre line that we used for most of our experiments shows some Cre expression in precerebellar nuclei at least at some stages of development, the differential effects of DREADD inhibition that we observed on the effect of cortical vs nuclear optogenetic stimulation argues against the possibility of direct inhibition of MFs in these experiments.

5. New figure 2:

- The normalization in panels C-D is somewhat confusing. The methods (lines 1301-2) describe normalizing within batches of pseudoconditioned mice, but it was unclear (to me) how these batches were defined.

To clarify the definition of ‘batches’, we have now added a separate paragraph in the Methods named “Mice for quantification of structural changes” (line 1238-1242), which reads: “Mice used for the quantification of VGLUT1, VGLUT2 and Gephyrin were conditioned or pseudo-conditioned in multiple batches. Each batch consisted of age and sex-balanced conditioned and pseudo-conditioned mice. All mice were subjected to 10 days of conditioning or pseudo-conditioning, where the latter group received CS and US at random rather than fixed ISIs, which prevented learning.”

To further clarify how we dealt with this in the analysis (e.g., data shown in panels C-D), we have now added the following sentences to the Methods section (line 1490-1493): “Histological procedures were performed at the same time for the pseudo-conditioned and conditioned mice within each batch of mice. To correct for unwanted differences between

batches of mice that were trained at different time points, values were normalized to the values of the pseudo-conditioned group for each batch of mice.”

It also seems appropriate to have the 100% tick labeled in such a normalized bar graph, since this is what we are comparing to.

We have now modified panels C and D to have the 100% tick on the y-axis.

- It is not clear from panels E-F that the x-axis is normalized. This should be clear.

For both panels we have now added “(norm)” to the x-axis label to indicate normalization.

- I personally find the y-labels in panels I and O (i.e. n/mm) to be confusing. Please make these labels more informative.

We have updated the y-axis label of panels I and O to improve clarity. The y-axis label of panel I now reads: “n/mm axon length” and of panel O now reads: “n/mm soma perimeter (norm)”.

6. Discussion of GrC stimulation experiments (lines 211-218): The authors use excitation and inhibition to mean increases and decreases in spiking. Please use the more accurate terms.

We thank the Reviewer for noticing this. We modified this sentence, which now reads as follows (line 232-236): “Most (i.e., 13 out of 15) of the recorded AIP neurons showed mean decreases rather than mean increases in spiking in response to gc-ChR2 stimulation (Supplementary Fig. 4c,d), indicating that activation of the MF - granule cell - PC pathway is unlikely to result in spike facilitation in CN neurons and that the facilitation outcomes highlighted above could indeed result from direct MF to CN neuron stimulation.”

7. The authors describe (lines 302-312) that Vm in some neurons correlates with eyelid movements, and it does not in others. Then they say that in the (pre-selected) correlated group, the population exhibits a particular set of encoding properties and other group does not. But of course, this was pre-destined by the selection criteria. Please rephrase.

We understand the Reviewer’s concern. The selection of neurons to be labeled as ‘CR-dependent’ or ‘CR-independent’ was done primarily using ROC analysis, where we compared Vm amplitudes on CR trials versus NR trials. However, for a few neurons there were insufficient CR or NR trials to be able to perform ROC analysis. In these neurons, we instead considered the relationship between Vm amplitude and the amplitude of the CR, in order to determine whether the occurrence of CRs coincided with increased Vm transients.

The subsequent correlation analysis is still relevant. Even though a preselection was done it would not necessarily be the case that Vm amplitude and area correlate with the size of the CR in these neurons. However, we agree with the Reviewer that it is highly likely that we would find a correlation there, given the small group size of CR-dependent neurons. The added value to the manuscript is this analysis is the finding that both the amplitude and the

area of the Vm response correlate with the CR, as well as data on the strength of the correlation.

*We modified the Results text to better reflect this point as follows (line 326-331):
“Furthermore, on a group level, both Vm amplitudes and area correlated with eyelid movements (Vm amplitude: $r = 0.435$, $P = 0.04$; area: $r = 0.545$, $P = 0.004$, Spearman, two-tailed, Bonferroni-corrected, $n = 34$ trials, $N = 3$ neurons; Supplementary Fig. 6b). As expected, for CR-independent neurons, Vm amplitudes and area did not correlate with eyelid movements (Vm amplitude: $r = 0.14$, $P = 0.189$; area: $r = 0.035$, $P = 1$, Spearman, two-tailed, Bonferroni-corrected, $n = 203$ trials, $N = 6$ neurons; Supplementary Fig. 6c).”*

To better clarify this point, we have also added the following text to the Methods section (line 1405-1409): “To classify neurons as CR-dependent or CR-independent, we performed ROC analysis on the above time periods, comparing CR vs. NR trials. For a few neurons there were insufficient CR or NR trials to be able to perform ROC analysis and in these neurons we instead considered the relationship between Vm amplitude and CR amplitude, in order to determine whether the occurrence of CRs coincided with increased Vm responses.”

8. The authors performs some truly heroic experiments in recording membrane dynamics during eye-blink conditioning (detailed in Figure 3-4). Only a subset of these neurons exhibit membrane dynamics that one would expect from the spike responses shown in previous studies, but they are actually in agreement with the prevalence of response types shown in this study (i.e. Figure S9). I think this is a point that is worth emphasizing a bit more.

We thank the Reviewer for pointing this out and agree that this finding could be made more explicitly. We have now added the following sentence to the Results paragraph discussing the juxtosomal patch data and Supplementary Fig. 9 (line 447-449): “These percentages are in good agreement with the distribution of CS-evoked Vm responses (Fig. 3g) as well as previously published data²².”

9. It seems to me that the analysis in Figure 5 is doing a disservice to the data that has been collected. I suspect that if the authors applied some sort of weighting function, rather than simply pooling, before applying LDA, they would get better performance of their decoder. Surely, this information exists within the population (e.g. Figures 4B, S6B, S9C, among others).

We appreciate the Reviewer's observation and concern regarding the potential underestimation of decoding accuracy due to the absence of a weighting factor in our analysis, and we would like to further elucidate our rationale for this choice.

Following the helpful and constructive Reviewer comments we received from our last revision of this manuscript, we decided to perform LDA on the entirety of the dataset, thereby avoiding making any preselections or assumptions that might influence the analysis. This decision was guided by our desire to preserve the raw dataset and to allow the algorithm to extract patterns and information directly from the observed neuronal responses as we recorded them. It is essential to highlight that while decoding accuracy is a critical parameter of interest, it is not the sole focus. We also consider the temporal dimension as a vital aspect.

The relation between decoding accuracy and time provides valuable insights into when during a trial relevant information is encoded by the neural population. This aspect may change when applying weighting factors or preselections.

We would like to emphasize that the timing of significant decoding accuracy coincides with decoding based on spike activity, which is a noteworthy finding. This convergence of results between different decoding suggests that the temporal aspects of information encoding are consistent across multiple analysis methods. While we acknowledge the potential advantages of applying weighting factors for decoding accuracy, we believe that our choice to perform LDA without them is justified.

10. In the discussion, the authors imply that their chemogenetic experiments result in a complete remove of the cerebellar cortex's influence. This is not strictly true – e.g. Purkinje cells probably still get climbing fiber input and fire complex spikes in response to the US. The authors should qualify accordingly.

We agree and we thank the Reviewer for pointing this out. We have modified the first paragraph of the discussion to more correctly reflect this. We changed “Chemogenetically silencing” to “Chemogenetic inhibition of cerebellar granule cells”, and lines 533-535 now read: “Taken together, these results strongly suggest that there is a capacity for well-timed learning within the AIP, without the need for MFs to convey CS information to the cerebellar cortex.”

Reviewer #2 (Remarks to the Author):

The thoroughly revised manuscript includes many new experiments and data points that significantly strengthen the conclusions drawn. Specifically the new direct optogenetic stimulation of mossy fiber terminals in the AIP closes an important gap that was apparent in the first version of the manuscript, and the added anatomical data result in a more solidified and bilateral finding of mossy fiber terminal changes in the AIP due to conditioning. The data on which the conclusions rest now look sound. Further, the removal of using EPSP/IPSP verbiage makes the interpretation of whole cell recordings more clear and commensurate with the measurements taken. The conclusions now drawn clearly add some important insights to mouse cerebellar eyeblink conditioning with respect to the involvement of mossy fiber input to the AIP. They also illustrate the complexity of evoked movements both in naïve and in trained mice in a way that is helpful to further development of this field.

We thank the Reviewer for their positive comments.

I only have a few remaining minor questions / comments.

Minor Comments.

The optogenetic CS for 200 or 500 ms eyeblink conditioning lasts throughout the entire delay period unlike a classic short sensory CS (Fig. S1). The authors should make this difference explicit in their interpretation of results with respect to what it means for the

control of learning the timing of a CS-US interval. Would the mice have learned a 500 ms CS-US pairing if the CS similarly only lasted 200 ms as in the 200 ms pairing?

*Thank you for this question and we apologize for any confusion. In all of the experiments in this paper we are concerned with only cerebellum-dependent **delay** eyeblink conditioning, in which the CS and the US always co-terminate. Trace conditioning, in which there is a gap between the CS and the US, requires not only an intact cerebellum but also includes contributions from other brain areas such as hippocampus and medial prefrontal cortex. Our data do not speak to the role of the cerebellar nuclei in trace conditioning. We have now clarified throughout the manuscript that we are focusing on cerebellum-dependent delay conditioning, in which the CS and US always co-terminate.*

Lines 164/165. 1292-1310. Please state if the investigators were blinded as to which type of animal was analyzed. Without blinding it is not clear if this method of scoring varicosities does not allow for a subjective factor, like unconscious bias.

To make it more clear that all quantifications were done with the observer blind to the experimental group analyzed, we added the following sentence (line 151-152): “All quantifications to investigate structural changes during learning were done by an investigator blind to the experimental group.”

Please note that we also detail this in the Methods section (line 1477-1478): “Quantitative evaluations were made by a blind experimenter using Image J software.”

Lines 474-477; 488/489. It seems tenuous to attribute the US responses in the AIP to CF input. There would be also mossy fiber input, including sensory feedback elicited by the movement. For a close association with CF input, the relative delay of CF input and depolarization should be shown to match. Does CF input also show a preceding CR influence like the Vm encoding of the US? Otherwise the Vm encoding would not seem to be a direct outcome of CF activation.

The Reviewer highlights an important point. Our assumption that the CF pathway is mainly responsible for the US-evoked Vm responses in AIP is based on a body of studies underlining the role of the CF pathway in conveying US information. However, we cannot exclude the possibility that US information also arrives in the AIP through the MF pathway. In fact, in the light of evolving models of the neural circuitry underlying EBC, it is highly likely that this is the case. For example, CS information is also conveyed by the US pathway, evident as Purkinje cell complex spikes after the CS (at the start of CRs), that induce short pauses in a subset of cerebellar nuclei neurons (Ten Brinke et al., 2015, 2017; Ohmae & Medina, 2015). While we did not see evidence of CS-evoked hyperpolarizations that could represent PC complex spikes at a similar time interval in our Vm traces, this does indicate that the stimuli are likely traveling along multiple pathways. In this study we did not record CF or MF activity during the US specifically, so a causative analysis will not be possible.

To address the concern expressed by the Reviewer, we have now removed the following sentence from the Results: “The US is thought to mainly activate neurons in the inferior olive,

which leads to complex spikes in PCs through CF activity⁵⁰, which also have collaterals that innervate the CN^{51,52}.”

Reviewer #3 (Remarks to the Author):

In this revised manuscript, the authors have addressed most of the initial critiques by all reviewers and added an impressive amount of new data and experiments. As a result, it is a much stronger paper. In particular, the inclusion of the direct mossy fiber stimulation in the AIP and the corresponding finding regarding timing of MF-AIP CRs is interesting and challenges a long-standing model in the field, which greatly increases the significance. The restructuring of the paper as a result of the newly added experiments as well as the inclusion of a larger number of neurons for the whole cell patch clamp experiments alleviate my concerns about the study being under-powered.

We thank the Reviewer for their positive comments.

I have no further major concerns. However, if the manuscript is accepted, I urge the authors to double-check all references before submitting the final version as I again noticed many apparently incorrect references. I cannot provide an exhaustive list but as an example most of the references in the third paragraph of the Introduction do not appear to be correct.

We apologize for the mistake in the references. We have carefully checked all references and made sure that they are correct.

REVIEWERS' COMMENTS

Reviewer #1 (Remarks to the Author):

The authors have addressed my concerns, so I have no further objections to the publication of the study.

REVIEWERS' COMMENTS

Reviewer #1 (Remarks to the Author):

The authors have addressed my concerns, so I have no further objections to the publication of the study.

We thank the reviewer.